# Hybrid-Vlasov modelling of nightside auroral proton precipitation during southward interplanetary magnetic field conditions

Maxime Grandin[1], Markus Battarbee[1], Adnane Osmane[1], Urs Ganse[1], Yann Pfau-Kempf[1], Lucile Turc[1], Thiago Brito[1], Tuomas Koskela[1], Maxime Dubart[1], and Minna Palmroth[1,2]

[1]University of Helsinki, Department of Physics, Finland
[2]Finnish Meteorological Institute, Helsinki, Finland

**Correspondence:** M. Grandin (maxime.grandin@helsinki.fi)

**Abstract.** Particle precipitation plays a key role in the coupling of the terrestrial magnetosphere and ionosphere by modifying the upper atmospheric conductivity and chemistry, driving field-aligned currents, and producing aurora. Yet, quantitative observations of precipitating fluxes are limited, since ground-based instruments can only provide indirect measurements of precipitation while particle telescopes onboard spacecraft merely enable point-like in-situ observations with inherently coarse time resolution above a given location. Further, orbit time scales generally prevent the analysis of whole events. On the other hand, global magnetospheric simulations can provide estimations of particle precipitation with a global view and higher time resolution. We present the first results of auroral ($\sim$1–30 keV) proton precipitation estimation using the Vlasiator global hybrid-Vlasov model in a noon-midnight meridional plane simulation driven by steady solar wind with southward interplanetary magnetic field. We first calculate the bounce loss cone angle value at selected locations in the simulated nightside magnetosphere. Then, using the velocity distribution function representation of the proton population at those selected points, we study the population inside the loss cone. This enables the estimation of differential precipitating number fluxes as would be measured by a particle detector onboard a low-Earth-orbiting spacecraft. The obtained differential flux values are in agreement with a well-established empirical model in the midnight sector, as are also the integral energy flux and mean precipitating energy. We discuss the time evolution of the precipitation parameters derived in this manner in the global context of nightside magnetospheric activity in this simulation, and we find in particular that precipitation bursts of $< 1$ min duration can be self-consistently and unambiguously associated with dipolarising flux bundles generated by tail reconnection. We also find that the transition region seems to partly regulate the transmission of precipitating protons to the inner magnetosphere, suggesting it has an active role in regulating ionospheric precipitation.

## 1 Introduction

The terrestrial atmosphere and ionosphere are known to be affected by the precipitation of particles coming from the magnetosphere and the solar wind. Precipitating protons in the keV energy range produce diffuse auroral emission, principally in the

auroral oval and in the cusp region (Hardy et al., 1989). Some emission lines are only produced during proton aurora, namely the $H_\alpha$ and $H_\beta$ lines (at 656.3 nm and 486.1 nm, respectively), and the Lyman-$\alpha$ line at 121.7 nm (Lummerzheim et al., 2001). It is hence possible to distinguish aurora related to proton precipitation from the aurora produced by electron precipitation.

Contrary to precipitating electrons which may be significantly affected by parallel electric fields in the auroral acceleration region (e.g., Lin and Hoffman, 1982), the acceleration (or deceleration) of auroral-energy precipitating protons by parallel electric fields resulting from a field-aligned potential drop is negligible (Liang et al., 2013). This enables a mapping of the proton aurora in the ionosphere to the magnetospheric region from which the particles originate. Frey et al. (2003) used this property to produce evidence of continuous magnetic reconnection at the magnetopause, as a proton aurora spot was observed to persist for several hours in the cusp region.

Within the auroral region, proton precipitation maximises in the pre-midnight sector in terms of total energy flux, but the mean precipitating energy peaks in the evening sector near 18 magnetic local time (MLT), where it can exceed 20 keV (Galand et al., 2001). The most common form of nightside proton aurora is a diffuse arc equatorwards from the electron aurora in the pre-midnight sector (Hardy et al., 1989). In addition, Nomura et al. (2016) have presented recent observations of pulsating proton aurora patches in the evening sector.

On the nightside, precipitating protons mostly originate from the central plasma sheet (e.g., Eather, 1967; Gilson et al., 2012; Spanswick et al., 2017). Proton precipitation is most commonly associated to two mechanisms. First, pitch-angle scattering into the bounce loss cone can take place in regions where the magnetic field curvature radius $R_c$ is within the same order of magnitude as the gyroradius $r_L$ of particles, as can be the case near the neutral sheet (Tsyganenko, 1982; Sergeev and Tsyganenko, 1982). This criterion has been formalised as $0 < \kappa \leq \sqrt{8}$, with $\kappa = \sqrt{R_c/r_L}$ (Sergeev et al., 1983). Second, loss-cone scattering of protons can be due to wave–particle interactions. Electromagnetic ion cyclotron (EMIC) waves are the prime candidate for such interactions (e.g., Erlandson and Ukhorskiy, 2001; Sakaguchi et al., 2008; Popova et al., 2018), but more recently, Xiao et al. (2014) suggested that fast magnetosonic waves, also known as equatorial noise, might interact with protons across a broad range of magnetic local times. It has been found, for instance, that pulsating proton aurora can be produced by non-linear interactions with EMIC waves, which show as Pc1 pulsations in ground-based magnetometers (Ozaki et al., 2016). On the dayside, it has been found that proton aurora flashes can be observed equatorwards from the cusp in relation with EMIC waves associated with plasma pressure pulses in the magnetosphere (Yahnina et al., 2008). More generally, EMIC waves can be responsible for auroral proton precipitation equatorwards of the proton oval forming long-lasting spots, arcs and flashes (Yahnin et al., 2018). Besides pitch-angle scattering associated with the $\kappa$ parameter and with EMIC waves, protons may precipitate because of configurational changes and magnetic reconnection (Mende et al., 2002).

Observations of proton precipitation can be achieved from space using particle detectors onboard spacecraft or from the ground using optical instruments or radars. Low-Earth-orbit (LEO) satellites comprising particle detectors, such as those of the Defense Meteorological Satellite Programme (DMSP) or NOAA 6, have been gathering electron and proton data in the keV energy range for several decades (Galand, 2001). Other spacecraft, such as the NASA Thermosphere Ionosphere Mesosphere Energetics and Dynamics (TIMED) or Imager for Magnetosphere-to-Aurora Global Exploration (IMAGE), have been measur-

ing auroral emission from space, in particular the hydrogen lines in the ultraviolet range (e.g., Zhang et al., 2005; Frey et al., 2002; Hubert et al., 2003).

From the ground, optical instruments such as meridian scanning photometers or all-sky imagers can be used to monitor proton precipitation by measuring the auroral emission at the hydrogen lines. Recently, two Forty-Eight Sixty-One (FESO) meridian scanning photometers, designed to observe the second hydrogen Balmer line (486.1 nm), have been deployed in Canada to provide observations of the proton aurora with enhanced sensitivity and sample rates (Unick et al., 2017). Ground-based optical instruments can be used in conjunction with satellite observations (e.g., Donovan et al., 2008). Besides, since energetic proton precipitation produces ionisation enhancements in the $E$ region of the ionosphere, ionospheric radars such as incoherent scatter radars can provide indirect observations of proton precipitation (e.g., Lyons et al., 2010).

In numerical studies, two aspects of proton precipitation can be distinguished. First, empirical models describing precipitation fluxes and locations have been developed. One example of these is the Hardy model (Hardy et al., 1989, 1991) which provides the average precipitation patterns as a function of the Kp index obtained by compiling two years of DMSP spacecraft data. This statistical model describes proton precipitation in a spatial grid containing 30 bins in corrected geomagnetic latitude above $50°$ and 48 bins in MLT, at seven levels of geomagnetic activity from $\mathrm{Kp} = 0$ to $\mathrm{Kp} \geq 6-$. The model gives the proton precipitation differential number flux in the 20 DMSP energy channels (from 30 eV to 30 keV with regular spacing in logarithmic scale) as well as the integral number flux, the integral energy flux and the mean precipitating energy. The Hardy model has been used as an input in a number of ionospheric models, e.g., Transcar and IPIM (Blelly et al., 2005; Marchaudon and Blelly, 2015). More recently, the OVATION Prime model (Newell et al., 2014), derived from a combination of DMSP particle data and global ultraviolet imager (GUVI) data from the TIMED spacecraft, has been developed and includes proton precipitation. However, due to the lack of DMSP satellites orbiting in the postmidnight sector, the magnetic local time (MLT) sectors comprised between 00:15 and 03:30 MLT are described using a linear interpolation.

Second, models based on first principles focus essentially on ionospheric effects of precipitating ions, such as the linear transport model described in Basu et al. (2001). In such approaches, precipitating fluxes are taken as model inputs, and the transport code provides ion production rates and auroral emission intensity profiles in the ionosphere. From a magnetospheric point of view, while simulations of electron precipitation have been achieved with various models (e.g., Palmroth et al., 2006; Raeder et al., 2008), attempts to model proton precipitation are scarce. A few magnetohydrodynamics (MHD) models have addressed the issue of relating particle precipitation characteristics to the magnetotail dynamics (Gilson et al., 2012; Ge et al., 2012, see below); however, currently no global kinetic simulations of the near-Earth environment have undertaken this task.

One obvious aspect through which proton precipitation and magnetotail dynamics can be tied is the relationship between auroral streamers and bursty bulk flows (BBFs). Bursty bulk flows are fast plasma flows propagating in the magnetotail along the Sun–Earth direction, either earthwards or away from the Earth (Angelopoulos et al., 1992, 1994). They are also characterised by an enhanced $B_z$ (northward component of magnetic field) and reduced plasma pressure (e.g., Ohtani et al., 2004). A given BBF may embed several dipolarising flux bundles (DFBs), which are themselves coherent magnetic structures exhibiting a $B_z$ increase by up to a few tens of nanoteslas (Liu et al., 2013) and may play a role in particle energisation in the magnetotail (Runov et al., 2017). Auroral streamers are thin auroral structures oriented in the north–south direction (e.g., Nishimura et al.,

2011). They have been associated with BBFs based on observational evidence combining spacecraft data with ground-based auroral imager, magnetometer and coherent scatter radar data (e.g., Fairfield et al., 1999; Nakamura et al., 2001a, b; Amm and Kauristie, 2002; Sergeev et al., 2004; Gallardo-Lacourt et al., 2014). On modelling aspects, Ge et al. (2012) reproduced proton precipitation enhancements associated with approaching BBFs with a global MHD model and found good agreement with spacecraft and ground-based auroral observations.

In this paper, we present an overview of nightside proton precipitation in a global magnetospheric hybrid-Vlasov simulation under southward interplanetary magnetic field (IMF) conditions, using the Vlasiator model (von Alfthan et al., 2014; Palmroth et al., 2018). Contrary to previous work in which the main focus was on the ionosphere, we here look at precipitation from the perspective of the magnetosphere and unambiguously tie the precipitation characteristics to the self-consistent simulation results. In particular, following the study by Juusola et al. (2018a) investigating flow bursts inside a BBF in the same Vlasiator run, we examine how proton precipitation can be related to dipolarising flux bundles. Section 2 describes the Vlasiator model as well as the methods developed to estimate proton precipitation from its outputs. Section 3 presents the results of the study, i.e., the features of nightside proton precipitation in the simulation under southward IMF. Section 4 discusses those results, and sect. 5 summarises the main conclusions.

## 2  Methods

### 2.1  Vlasiator

This study relies on numerical simulations using Vlasiator, a global hybrid-Vlasov model of the near-Earth plasma environment (von Alfthan et al., 2014; Palmroth et al., 2018). In the hybrid-Vlasov approach, protons are described as velocity distribution functions (VDFs) in a grid of the phase space (ordinary space and velocity space), whereas electrons are treated as a massless, charge-neutralising fluid. The time evolution of the VDFs is obtained by solving the Vlasov equation, and closure of the system is achieved with Ohm's law including the Hall term (Palmroth et al., 2018).

The simulation run used in this study is a 2D-3V (two-dimensional in ordinary space, three-dimensional in velocity space) simulation in the noon-midnight meridional plane of the magnetosphere, i.e., $XZ$ in the geocentric solar magnetospheric (GSM) coordinate system. The origin of the simulation domain is set in the Earth's centre. The simulation box in ordinary space spans from $X = -94 R_E$ (Earth radii, $1 R_E = 6371\,\mathrm{km}$) on the nightside to $X = +47 R_E$ on the dayside, and is comprised between boundaries at $Z = \pm 57 R_E$ in the north-south direction, with a grid resolution of 300 km. The inner boundary of the magnetospheric domain lies at $\sim 4.7 R_E$ from the center of the Earth. In each cell of the ordinary space, a velocity grid extends between $\pm 4020\,\mathrm{km\,s^{-1}}$ with a resolution of $30\,\mathrm{km\,s^{-1}}$ in $V_X$, $V_Y$, and $V_Z$. A sparsity threshold is implemented in Vlasiator, below which phase-space density values are discarded to limit the computational load of the simulation. In the run used in this study, the sparsity threshold is set to $10^{-15}\,\mathrm{m^{-6}\,s^3}$.

The near-Earth environment is driven by incoming solar wind from the $+X$ wall of the simulation domain. The solar wind population consists of protons characterised by Maxwellian VDFs with a number density of $1\ \mathrm{cm^{-3}}$, a temperature of 500 kK, a bulk velocity of $-750\,\mathrm{km\,s^{-1}}$ along the $X$ axis, a dynamic pressure of about 0.9 nPa, and carrying a purely

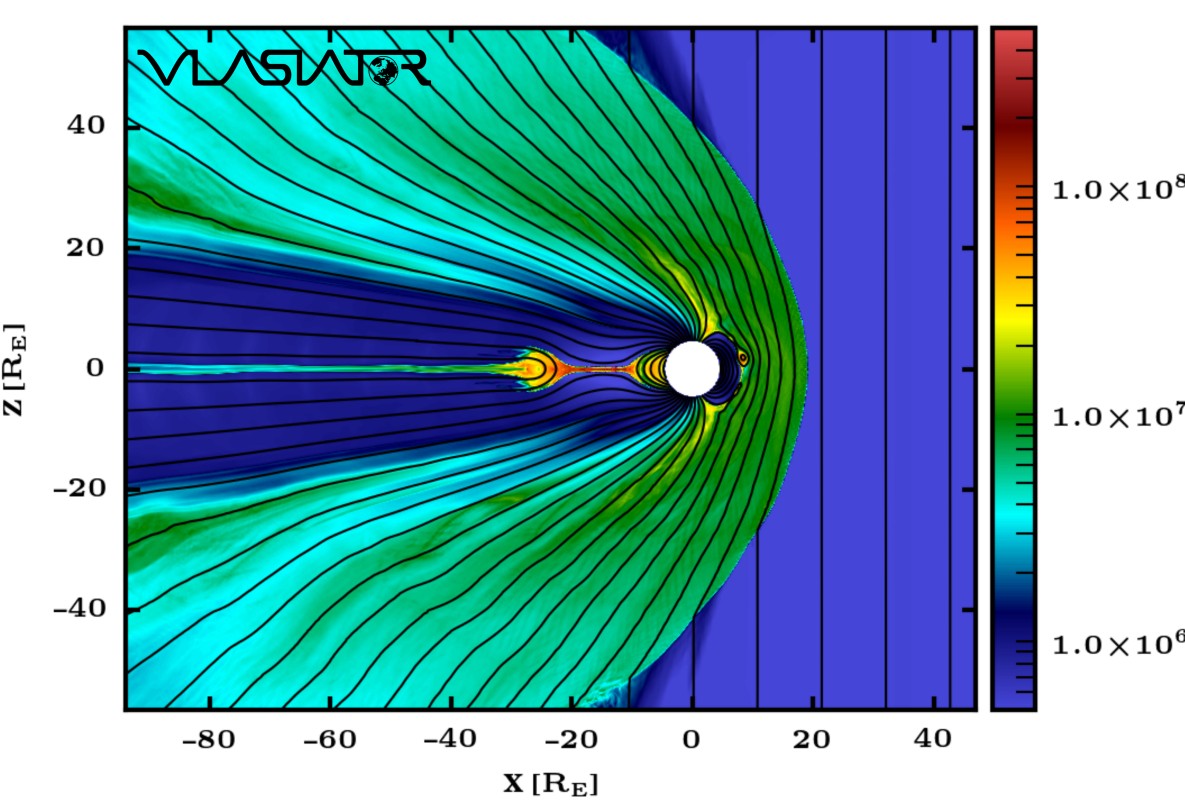

**Figure 1.** Simulated plasma temperature in the entire simulation domain, at $t = 1800$ s. The white area corresponds to near-Earth space located inside the inner boundary at 4.7 $R_E$. Black lines represent magnetic field lines.

southward interplanetary magnetic field (IMF) with a magnitude of 5 nT. Other boundary conditions are as follows: von Neumann condition for the $-X$ and $\pm Z$ walls, periodic conditions in the out-of-plane direction ($\pm Y$), whereas the inner boundary is a perfectly conducting cylinder with a static Maxwellian VDF for protons. Finally, the geomagnetic field is given by a 2D line dipole along the $Z$ axis, centred at the origin, and scaled to obtain a realistic magnetopause standoff distance
5   (Daldorff et al., 2014).

In the solar wind, the ion inertial length is $\lambda_p = 228$ km, and the proton Larmor radius is $r_L = 214$ km. A dedicated study by Pfau-Kempf et al. (2018) showed that the ordinary space resolution of 300 km is sufficient to resolve most of the proton kinetics in such conditions. Besides, the Alfvén speed is $V_A = 109$ km s$^{-1}$, which is significantly greater than the velocity space resolution of 30 km s$^{-1}$. Similarly, the ordinary space and velocity space resolutions are sufficient to resolve the ion
10  kinetics in the transition region between stretched and dipole-like magnetic field lines in the magnetotail, where $\lambda_p = 906$ km, $r_L = 681$ km and $V_A = 1596$ km s$^{-1}$.

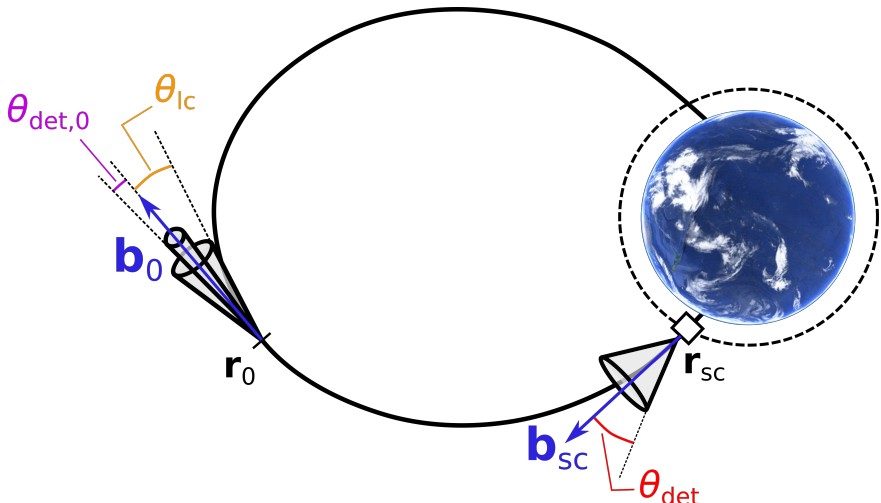

**Figure 2.** Geometry used in the derivation of directional differential flux of precipitating protons.

Figure 1 gives an overview of the simulated area at a time step ($t = 1800\,\mathrm{s}$) when nightside reconnection is taking place. The plotted parameter is the plasma temperature, and the black lines correspond to magnetic field lines. The incoming solar wind at 500 kK with southward IMF can be seen on the right-hand side of the figure, and the major geospace regions (bow shock, magnetosheath, dayside magnetopause, polar cusps, magnetotail lobes, plasma sheet) can be identified. A plasmoid is forming

in the nightside magnetosphere (Palmroth et al., 2017), whose signature is visible between $X = -30R_E$ and $X = -20R_E$. Juusola et al. (2018b) studied the plasma sheet during this same run and evaluated its thickness to lie essentially within 0.1– 0.2 $R_E$ (i.e., just a few simulation cells) in the region of interest for our study ($X > -20R_E$).

This simulation run was previously used in several studies focusing on various phenomena and regions in the near-Earth space, such as dayside magnetopause reconnection (Hoilijoki et al., 2017), ion acceleration in the magnetosheath (Jarvinen

et al., 2018), magnetotail reconnection (Palmroth et al., 2017; Juusola et al., 2018a), and magnetotail current sheet flapping (Juusola et al., 2018b). In each of these studies, the presented results showed good correspondence with earlier findings. Here, this run will be used to characterise nightside auroral proton precipitation in the simulation by making use of the VDF description of the proton population.

### 2.2 Precipitation spectra calculation

Figure 2 illustrates the definitions of angles and vectors used in this section. The configuration is shown for particles precipitating in the southern hemisphere to be consistent with the results shown in the rest of the paper. For particles precipitating in the northern hemisphere, the configuration is symmetrical.

At a given three-dimensional location **r** of the ordinary space, the differential intensity $\mathcal{J}$ (more generally called directional differential flux) of protons at energy $E$ for the direction along unit vector **u** is related to the velocity distribution function $f$

by (see 6.4.1. in Baumjohann and Treumann, 1997)

$$\mathcal{J}(E, \mathbf{u}, \mathbf{r}) = \frac{v^2}{m_p} f(\mathbf{r}, v, \mathbf{u}), \tag{1}$$

where $m_p$ is the proton mass and $v = \sqrt{2E/m_p}$ is the velocity magnitude. $\mathcal{J}$, when expressed in SI units, is given in particles $\mathrm{m}^{-2} \, \mathrm{s}^{-1} \, \mathrm{sr}^{-1} \, \mathrm{J}^{-1}$.

Particle detectors onboard spacecraft estimate $\mathcal{J}$ by taking the average of the incoming particle flux in a given energy channel over the detector's surface $S_{\mathrm{det}}$ and viewing angle $\Omega_{\mathrm{det}}$, which can be written

$$\hat{\mathcal{J}}(E, \mathbf{n}_{\mathrm{det}}, \mathbf{r}_{\mathrm{sc}}) = \frac{1}{\int_{S_{\mathrm{det}}} \int_{\Omega_{\mathrm{det}}} \mathbf{u}' \cdot \mathbf{n}_{\mathrm{det}} \, \mathrm{d}\Omega \, \mathrm{d}S} \int_{S_{\mathrm{det}}} \int_{\Omega_{\mathrm{det}}} \frac{v^2}{m_p} f(\mathbf{r}_{\mathrm{sc}}, v, \mathbf{u}') \, \mathbf{u}' \cdot \mathbf{n}_{\mathrm{det}} \, \mathrm{d}\Omega \, \mathrm{d}S, \tag{2}$$

where $\mathbf{u}'$ is the unit vector of a given velocity direction inside $\Omega_{\mathrm{det}}$, $\mathbf{n}_{\mathrm{det}}$ is the unit vector normal to the detector's surface, and $\mathbf{r}_{\mathrm{sc}}$ is the location of the spacecraft, for instance near the top of the ionosphere. Strictly speaking, $\hat{\mathcal{J}}$ is a differential

intensity measured within a fixed solid angle and detector area. In practice, telescopes onboard spacecraft generally collect particles within a cone with an opening angle $\theta_{\mathrm{det}}$ of the order of $15°$ (e.g., Rodger et al., 2010). Assuming that the directional differential flux is uniform over the detector's surface and that $\mathbf{n}_{\mathrm{det}}$ is aligned with the local geomagnetic field direction $\mathbf{b}_{sc}$ (i.e., the telescope measures precipitating particles), and expressing the velocity distribution function in spherical coordinates $(v, \theta, \varphi)$ in the local magnetic frame, one gets

$$\hat{\mathcal{J}}(E, \mathbf{b}_{\mathrm{sc}}, \mathbf{r}_{\mathrm{sc}}) = \frac{1}{\int_0^{2\pi} \int_0^{\theta_{\mathrm{det}}} \cos\theta \sin\theta \, \mathrm{d}\theta \, \mathrm{d}\varphi} \int_0^{2\pi} \int_0^{\theta_{\mathrm{det}}} \frac{v^2}{m_p} f(\mathbf{r}_{\mathrm{sc}}, v, \theta, \varphi) \cos\theta \sin\theta \, \mathrm{d}\theta \, \mathrm{d}\varphi, \tag{3}$$

or, by defining $\mu = \cos\theta$,

$$\hat{\mathcal{J}}(E, \mathbf{b}_{\mathrm{sc}}, \mathbf{r}_{\mathrm{sc}}) = \frac{v^2}{m_p} \frac{1}{\int_0^{2\pi} \int_{\mu_{\mathrm{det}}}^{1} \mu \, \mathrm{d}\mu \, \mathrm{d}\varphi} \int_0^{2\pi} \int_{\mu_{\mathrm{det}}}^{1} f(\mathbf{r}_{\mathrm{sc}}, v, \mu, \varphi) \mu \, \mathrm{d}\mu \, \mathrm{d}\varphi$$

$$= \frac{v^2}{m_p} \frac{2}{2\pi(1 - \mu_{\mathrm{det}}^2)} \int_0^{2\pi} \int_{\mu_{\mathrm{det}}}^{1} f(\mathbf{r}_{\mathrm{sc}}, v, \mu, \varphi) \mu \, \mathrm{d}\mu \, \mathrm{d}\varphi \tag{4}$$

with $\mu_{\mathrm{det}} = \cos\theta_{\mathrm{det}}$. In the symmetrical configuration where the spacecraft observes precipitating particles in the northern

hemisphere, $\mathbf{n}_{\mathrm{det}}$ is aligned with $-\mathbf{b}_{\mathrm{sc}}$, but angle definitions remain the same.

As a first-order approximation, protons remain attached to a given magnetic flux tube; one can hence trace particles backwards in time out to the magnetosphere, following magnetic field lines. Assuming smooth quasilinear propagation, for a given particle of pitch angle $\theta$, the quantity $\sin^2\theta/B$ is conserved along the trajectory. Let $\mathbf{r}_0$ be a location in the magnetosphere mapping to the spacecraft in terms of magnetic field topology. For particles travelling between $\mathbf{r}_0$ and $\mathbf{r}_{\mathrm{sc}}$, we have

$$\frac{\sin^2\theta_0}{B_0} = \frac{\sin^2\theta}{B_{\mathrm{sc}}}, \tag{5}$$

where $B_0$ and $\theta_0$ correspond to the magnetic field magnitude and the particle pitch angle at $\mathbf{r}_0$, respectively, and $B_{sc}$ and $\theta$ correspond to the same parameters at $\mathbf{r}_{sc}$. This can also be written

$$\frac{1-\mu_0^2}{B_0} = \frac{1-\mu^2}{B_{sc}}, \tag{6}$$

with $\mu_0 = \cos\theta_0$. By differentiating eq. (6), one gets

$$\mu\,\mathrm{d}\mu = \frac{B_{sc}}{B_0}\mu_0\,\mathrm{d}\mu_0. \tag{7}$$

According to Liouville's theorem, $f(\mathbf{r},v,\mu,\varphi)$ is conserved along the trajectories of the system, i.e., in the absence of potential fields

$$f(\mathbf{r}_{sc},v,\mu,\varphi) = f(\mathbf{r}_0,v,\mu_0,\varphi_0). \tag{8}$$

Using eq. (8), and then equations (6) and (7), yields from eq. (4)

$$\hat{\mathcal{J}}(E,\mathbf{b}_{sc},\mathbf{r}_{sc}) = \frac{v^2}{m_p}\frac{2}{2\pi(1-\mu_{\det}^2)}\int\limits_0^{2\pi}\int\limits_{\mu_{\det}}^1 f(\mathbf{r}_{sc},v,\mu,\varphi)\mu\,\mathrm{d}\mu\,\mathrm{d}\varphi$$

$$\hat{\mathcal{J}}(E,\mathbf{b}_{sc},\mathbf{r}_{sc}) = \frac{v^2}{m_p}\frac{2}{2\pi(1-\mu_{\det}^2)}\int\limits_0^{2\pi}\int\limits_{\mu_{\det}}^1 f(\mathbf{r}_0,v,\mu_0,\varphi_0)\mu\,\mathrm{d}\mu\,\mathrm{d}\varphi$$

$$\hat{\mathcal{J}}(E,\mathbf{b}_{sc},\mathbf{r}_{sc}) = \frac{v^2}{m_p}\frac{2}{2\pi(1-\mu_{\det,0}^2)\frac{B_{sc}}{B_0}}\int\limits_0^{2\pi}\int\limits_{\mu_{\det,0}}^1 f(\mathbf{r}_0,v,\mu_0,\varphi_0)\frac{B_{sc}}{B_0}\mu_0\,\mathrm{d}\mu_0\,\mathrm{d}\varphi_0$$

$$\hat{\mathcal{J}}(E,\mathbf{b}_{sc},\mathbf{r}_{sc}) = \frac{v^2}{m_p}\frac{1}{\int_0^{2\pi}\int_{\mu_{\det,0}}^1 \mu_0\,\mathrm{d}\mu_0\,\mathrm{d}\varphi_0}\int\limits_0^{2\pi}\int\limits_{\mu_{\det,0}}^1 f(\mathbf{r}_0,v,\mu_0,\varphi_0)\mu_0\,\mathrm{d}\mu_0\,\mathrm{d}\varphi_0 \tag{9}$$

$$\hat{\mathcal{J}}(E,\mathbf{b}_{sc},\mathbf{r}_{sc}) = \hat{\mathcal{J}}(E,\mathbf{b}_0,\mathbf{r}_0),$$

where $\mathbf{b}_0$ is the unit vector with the direction of the local magnetic field at $\mathbf{r}_0$, and

$$\mu_{\det,0} = \sqrt{1 - \frac{B_0}{B_{sc}}(1-\mu_{\det}^2)}. \tag{10}$$

In other words, the directional differential particle flux observed by the telescope onboard the spacecraft can be estimated by averaging the relevant subset of the particle VDF at a chosen conjugate location in the magnetosphere. The angular boundaries of the averaged phase space domain are obtained by scaling the viewing angle of the detector to $\theta_{\det,0} = \arccos\mu_{\det,0}$ based

on the ratio of magnetic field magnitudes at the spacecraft and at the conjugate location in the magnetosphere where the VDF is analysed (cf. eq. (10)).

For particles located at $\mathbf{r_0}$ to reach a spacecraft at the top of the ionosphere at $\sim$800 km altitude, their pitch angles must be within the bounce loss cone, whose opening angle is determined by

$$\theta_{lc} = \arcsin\sqrt{\frac{B_0}{B_{sc}}}. \tag{11}$$

In the undisturbed nightside equatorial plane, $\theta_{lc}$ takes values of a few degrees at most, and is smaller than $1°$ beyond $10\,R_E$ (see, e.g., Sergeev and Tsyganenko, 1982). In presence of a rather large dipolarisation front with, e.g., $B_0 \simeq B_z = 30$ nT, assuming a mapping to auroral latitudes ($B_{sc} \sim 53\,000$ nT), eq. (11) gives $\theta_{lc} = 1.4°$. In Vlasiator, knowing the VDF at a given location in the magnetosphere, one can calculate the bounce loss cone angle value and, at a given velocity magnitude $v$ (i.e.,

at a given energy), average the phase-space density inside the loss cone to evaluate $\hat{\mathcal{J}}(E, \mathbf{b}_0, \mathbf{r}_0)$. It should be noted that, with this approach, the obtained directional differential precipitating flux corresponds to the one which would be measured by a telescope onboard a spacecraft at the topside ionosphere with a viewing angle $\Omega_{det} = 2\pi\,\mathrm{sr}$ (cf. eq. (2)).

In practice, the methodology to estimate differential precipitating proton fluxes at $\mathbf{r}_0$ is as follows. First, the loss cone angle $\theta_{lc}$ is calculated. This is achieved by following the magnetic field line from $\mathbf{r}_0$ to the inner boundary of the simulation domain,

at about $4.7\,R_E$, and then extrapolate the magnetic field to the top of the ionosphere using the line dipole approximation. The ratio of the magnetic field magnitudes at $\mathbf{r}_0$ and at the mapped region in the topside ionosphere enables the calculation of the bounce loss cone angle using eq. (11). Second, we estimate the differential precipitating flux by calculating

$$\tilde{\mathcal{J}}(E, \mathbf{b}_0, \mathbf{r}_0) = \frac{v^2}{m_p} \langle f(\mathbf{r}_0, v, \theta, \varphi)\rangle_{\theta < \theta_0 = \theta_{lc}}, \tag{12}$$

where $\langle f(\mathbf{r}_0, v, \theta, \varphi)\rangle_{\theta < \theta_0}$ is the average value of the phase-space density at speed $v$ inside the bounce loss cone. This approx-

imation of eq. (9) is reasonable since $\theta_0$ is very small and hence we have $\mu_0 = \cos\theta_{lc} \approx 1$ inside the integral.

## 2.3 Examples of differential precipitating proton fluxes with Vlasiator

In the simulation used in this study, full velocity distributions of protons are saved every 50 simulation cells in the $X$ and $Z$ direction due to limitations in the file sizes. Therefore, the estimation of the differential precipitating proton flux based on eq. (12) can be applied only in a few selected locations in the nightside magnetosphere. Figure 3 shows examples of

velocity distributions observed at two virtual spacecraft $S_1$ and $S_2$ that are used in this study. The main panel (a) shows the proton temperature in the nightside part of the noon-midnight plane at time step $t = 1800\,\mathrm{s}$, with magnetic field lines drawn in black. It should be stressed that this temperature is that of an isotropic plasma which would have the same total pressure as the simulated distribution, thus it does not represent the temperature of the bulk plasma but rather the measured effect of a combined bulk plasma and potential fast additional flows. The effect of such a combination can be seen near $S_1$, as a

stream of hot ($T \approx 20$ MK) plasma coming from the transition region is reaching the virtual spacecraft, leading to a local enhancement of the proton temperature compared to its background value at $S_1$. The white area corresponds to the region of the geospace located earthwards from the inner boundary at $\sim 4.7R_E$. Black arrows indicate the magnetic field direction at the virtual spacecraft. Panels (b) and (c) show slices of the velocity distributions at $S_1$ and $S_2$, respectively, in the plane defined by the local magnetic field direction $\mathbf{v}_B$ and the local electric field direction $\mathbf{v}_{B\times V}$ (approximately aligned with the $Y$ direction,

i.e., into the simulation plane). The grid spacing is $1000$ km s$^{-1}$. The velocity distribution at $S_1$ consists of a core population centred near the origin and a crescent-shaped beam at about $1000$ km s$^{-1}$. It corresponds to the superposition of a cold plasma with a low bulk velocity in the magnetic field direction with a more energetic population travelling earthwards. The velocity distribution at $S_2$ is nearly Maxwellian with a broad loss cone in the magnetic field direction (this loss cone will be discussed

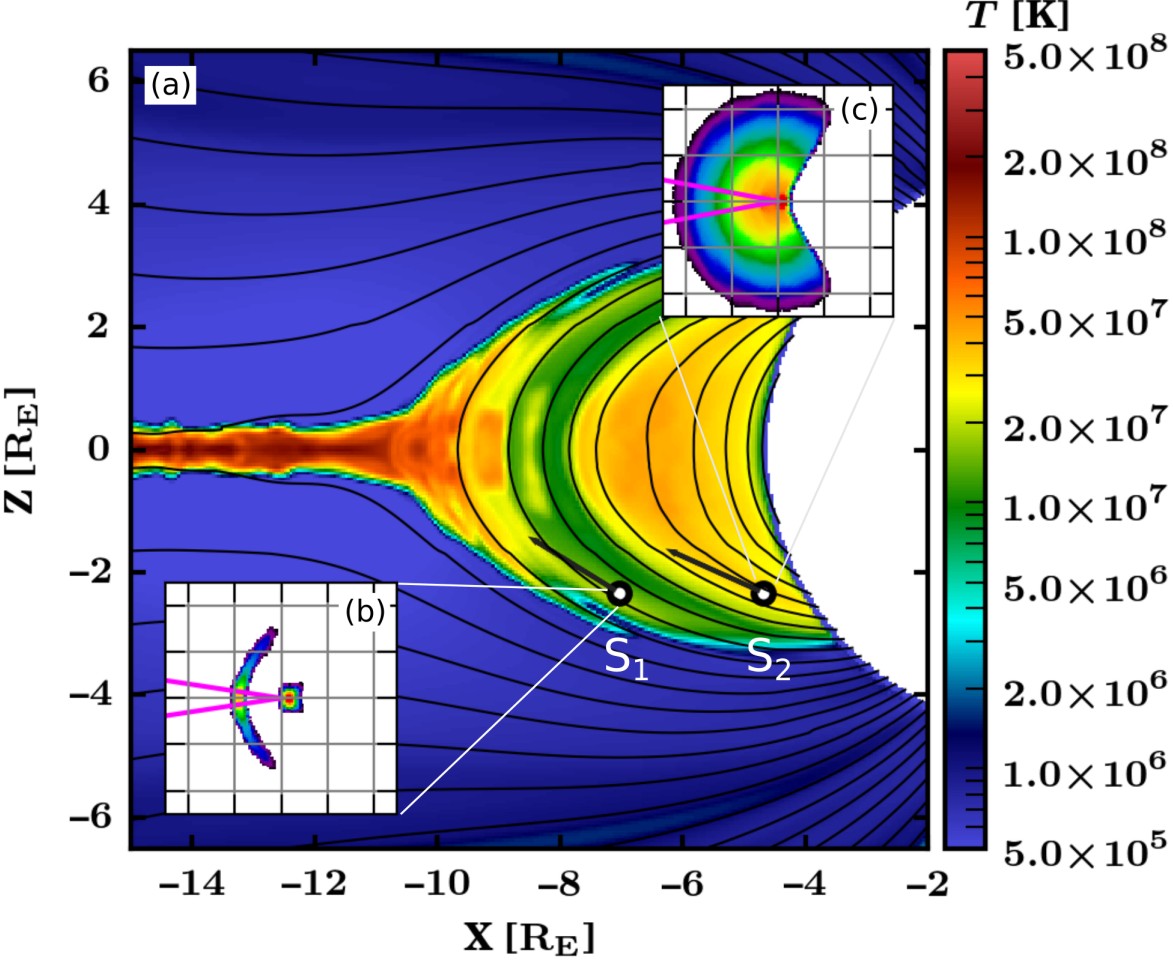

**Figure 3.** (a) Simulated plasma temperature in the nightside magnetosphere at $t = 1800\,\mathrm{s}$. The Sun is located beyond the right boundary of the figure. The two black circles filled with white are virtual spacecraft $S_1$ and $S_2$. The arrows indicate the magnetic field directions at $S_1$ and $S_2$. (b) Velocity distribution function of protons at $S_1$ in the $(v_\parallel, v_{B \times V})$ plane. The grid spacing is 1000 km s$^{-1}$. The magenta lines indicate the boundaries of the bounce loss cone. (c) Same for protons at $S_2$.

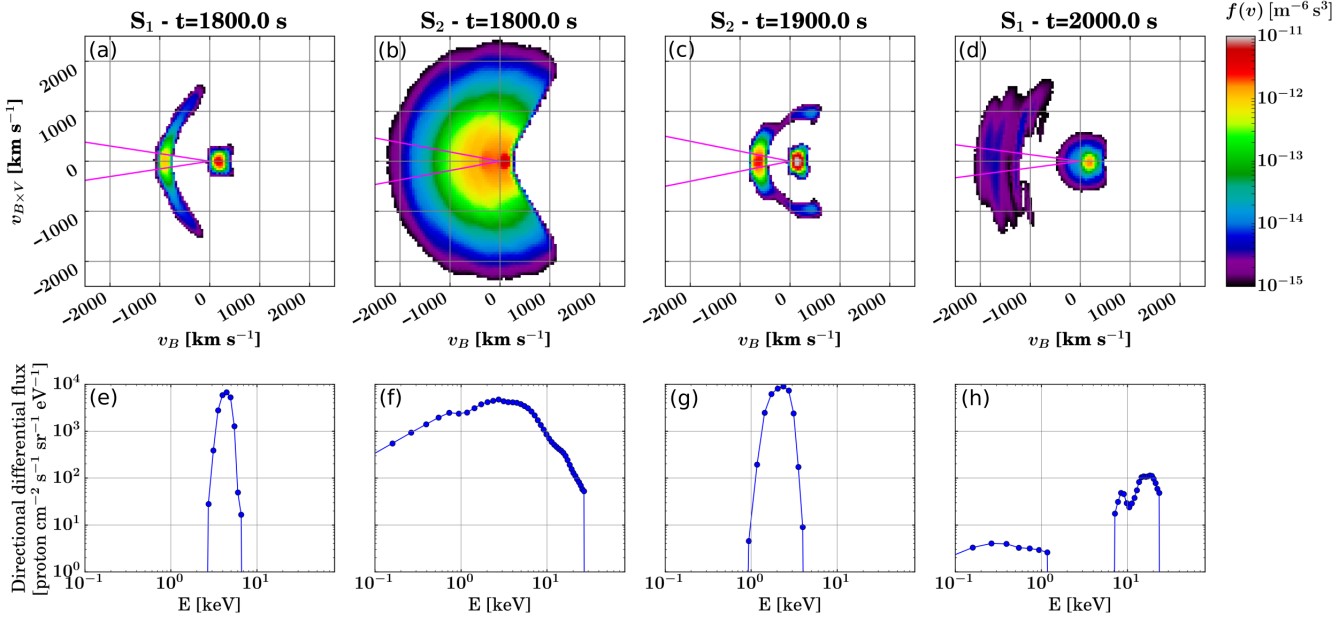

**Figure 4.** (top) Examples of proton velocity distribution slices in the local magnetic frame at virtual spacecraft $S_1$ and $S_2$ at various time steps of the simulation. The bounce loss cone is shown with magenta lines. (bottom) Corresponding directional differential precipitating fluxes obtained with the presented method.

below). The width of the distribution, centred near the origin, indicates that it corresponds to a hotter plasma (in comparison with the one at $S_1$) nearly at rest in the local magnetic frame, with protons having velocities up to about 2000 km s$^{-1}$. The imposed static Maxwellian VDF at the inner boundary can affect the neighbouring cells through the calculation of translation and acceleration terms in the Vlasov equation, but such effects vanish rapidly with distance to the boundary, and the plasma at

$S_2$ is unlikely to be affected by the boundary condition.

The contrast between the velocity distributions at $S_1$ and $S_2$ is reflected in the plasma temperature, which is 15 MK at $S_1$ and 32 MK at $S_2$. In panels (b) and (c), the bounce loss cone is indicated with pink lines. As can be seen in the two examples, the loss cone is not empty, and hence a directional differential flux of precipitating protons can be derived using eq. (12). We note that the empty part of the phase space in panel (c) is due to the presence of the inner boundary. Indeed, the inner boundary

absorbs incoming protons, and particles bouncing back from a mirror point earthwards from the inner boundary in the southern hemisphere are therefore absent at virtual spacecraft $S_2$ in the simulation. This feature does not have consequences in our study, since the relevant part of the velocity distribution function is the one located inside the bounce loss cone.

Figure 4 shows four examples of velocity distributions at virtual spacecraft $S_1$ and $S_2$ during the simulation run (panels a–d) and the associated precipitating proton differential fluxes obtained using the method described in section 2.2 (panels e–h).

The velocity distributions are slices in the plane defined by the magnetic field direction, $\mathbf{v}_B$, and the electric field direction, $\mathbf{v}_{B \times V}$. Panels (a) and (b) are the distributions at $t = 1800.0$ s at virtual spacecraft $S_1$ and $S_2$, respectively, shown in Fig. 3.

The crescent-shaped beam partly in the bounce loss cone shown in panel (a) is associated with a precipitating differential flux narrow in energies (panel e), peaking around 4–5 keV with values close to $10^4$ proton cm$^{-2}$ s$^{-1}$ sr$^{-1}$ eV$^{-1}$. Although this distribution is intrinsically unstable, the development of instabilities (magnetosonic, ion-ion left-hand, firehose) requires the conditions that the field-aligned drift velocity $V_0$ of the protons be greater than the Alfvén speed $V_A$, and that the ratio between beam ($n_b$) and core ($n_c$) densities be "very small" (Gary, 1991). While the distribution shown in panel (a) has $n_b/n_c = 0.086$, it exhibits $V_0 \sim 1000$ km s$^{-1}$ and $V_A = 4379$ km s$^{-1}$. Since the criterion on velocities is not verified, we do not expect that the instabilities listed above can grow fast enough to significantly affect the ion distributions on the time scale needed for precipitating protons (i.e., essentially the field-aligned beam) to reach the inner boundary or even the ionosphere.

In contrast with the distribution with a crescent-shaped beam shown in panel (a), the hot, nearly-Maxwellian velocity distribution inside the loss cone at S$_2$ (panel b) is associated to a broad precipitation spectrum (panel f), with energies of precipitating protons ranging from below 100 eV to nearly 30 keV. The peak differential flux values are of the order of 4000 proton cm$^{-2}$ s$^{-1}$ sr$^{-1}$ eV$^{-1}$, at energies of 2–5 keV. The sharp cutoff taking place at $\sim$25 keV in panel (f) is due to the sparsity threshold, set to $10^{-15}$ m$^{-6}$ s$^3$ in this run, hence corresponding to the lower limit of the colour axis in panels a–d. Panel (c) shows a velocity distribution relatively similar to panel (a) but was obtained at virtual spacecraft S$_2$ with a delay of 100 s, indicating that the region with narrow-beam precipitation moved earthwards. The corresponding differential flux in panel (g) is centred around 2–3 keV, i.e., slightly lower than in panel (e), while peak flux values are almost the same with nearly $10^4$ proton cm$^{-2}$ s$^{-1}$ sr$^{-1}$ eV$^{-1}$. The last velocity distribution, in panel (d), was taken at S$_1$ 200 s after the one from panel (a). The core population is broader and is partly inside the bounce loss cone, leading to a low flux of low-energy (0.1–1 keV) proton precipitation as can be seen in panel (h). A beam of low phase-space density with a complex structure partly fills the loss cone at velocities of 1300–2100 km s$^{-1}$, which shows in the precipitating spectrum as two peaks, at about 8 keV and 15 keV, with flux values two orders of magnitude lower than in panel (a). These four selected examples suggest that at virtual spacecraft S$_1$ and S$_2$ one can observe a variety of precipitating fluxes during the Vlasiator simulation. If compared to observations, these differential fluxes are within the same order of magnitude as the NOAA 6 proton energy spectra shown in Basu et al. (2001) in terms of peak energy (1–10 keV) and slightly higher in terms of flux values ($10^2$–$10^3$ cm$^{-2}$ s$^{-1}$ sr$^{-1}$ eV$^{-1}$). Observations from the DMSP 12 spacecraft reported in Lummerzheim et al. (2001) indicate mean precipitating energy of the order of 10 keV and flux values around $10^3$ cm$^{-2}$ s$^{-1}$ sr$^{-1}$ eV$^{-1}$. These values were however obtained in the evening MLT sector, where proton precipitation exhibits statistically harder energy spectra than in the midnight sector (Galand et al., 2001).

One assumption which is made during the derivation of the precipitating fluxes is that there are no field-aligned electric fields in the system (cf. eq. (8)). Figure S1 in the Supplementary Material shows the parallel component of the electric field at the same time step and with a similar format as Fig. 3. This parallel component was averaged over 120 s, which corresponds roughly to one bounce period for 10 keV protons at $L = 9$ (virtual spacecraft S$_1$). It can be seen that the parallel electric field between S$_1$ or S$_2$ and the inner boundary is of the order of 0.01 to 0.1 mV m$^{-1}$. When integrated along the field line between S$_1$ and the inner boundary, this corresponds to a potential difference of the order of 1 kV. In their discussion of the effect of potential drops in the auroral acceleration region on precipitating protons, Liang et al. (2013) estimate that for ions with energies $\gg 1$ keV the acceleration resulting from typical potential drops in the auroral acceleration region ($\sim 1$ kV up to

$\sim 4\,\mathrm{kV}$ occasionally) can be neglected, which enables a reasonable mapping of auroral latitudes to the central plasma sheet. It therefore seems acceptable to neglect the effect of parallel electric fields when deriving the precipitating proton fluxes with the method described in section 2.2. Furthermore, given the location of $S_1$ and $S_2$ on closed field lines which are being convected earthwards, there is no risk that the precipitating protons observed at $S_1$ (or $S_2$) are affected by processes such as magnetic reconnection before they reach the ionosphere.

## 3   Results

### 3.1   Nightside proton precipitation

An overview of the dynamics of the nightside magnetosphere from $t = 1000\,\mathrm{s}$ until the end of the simulation at $t = 2150\,\mathrm{s}$ is provided with supplementary animation S1. Figure 5 shows the time evolution of the proton precipitation at virtual spacecraft $S_1$ and $S_2$ during this time interval. Panels (a) and (b) show the precipitating proton differential flux (colour scale) as well as the mean precipitating energy (black line) as a function of time, while panels (c) and (d) show the integral energy flux. It can be seen that, at virtual spacecraft $S_1$, broad-energy proton precipitation above 1 keV starts around $t = 1360\,\mathrm{s}$, as the magnetic field line observed by $S_1$ is becoming slightly stretched. The energetic tail of the proton precipitation spectrum reaches up to 20 keV and the mean precipitating energy fluctuates between 2 and 5 keV until $t \approx 1750\,\mathrm{s}$, i.e., about 90 s after the global magnetotail reconfiguration is initiated by the dominant X-line near $X = -13R_E$. Around $t = 1750\,\mathrm{s}$, the precipitation becomes narrower in energy and, after $t = 1850\,\mathrm{s}$, the mean precipitating energy increases from 4 keV to 20 keV, while fluxes decrease by almost two orders of magnitude. This corresponds to the times during which the nightside reconnection speeds up and $S_1$ observes magnetic field lines becoming more dipolar. Between $t = 1900\,\mathrm{s}$ and $t = 1925\,\mathrm{s}$, $S_1$ does not observe any precipitation, as the virtual spacecraft is momentarily observing lobe-type plasma and the local loss cone is empty, but precipitation resumes with high energy (10–20 keV) and relatively low differential flux values ($\sim 10^2\,\mathrm{cm}^{-2}\,\mathrm{s}^{-1}\,\mathrm{sr}^{-1}\,\mathrm{eV}^{-1}$) after $t = 1925\,\mathrm{s}$, as $S_1$ is magnetically connected to the transition region. A low-energy ($< 1\,\mathrm{keV}$) precipitating population appears between $t = 1990\,\mathrm{s}$ and $t = 2040\,\mathrm{s}$, presumably associated with heating of the core plasma. At the end of the simulation, the precipitating mean energy decreases while differential flux values are enhanced to $\sim 10^3\,\mathrm{cm}^{-2}\,\mathrm{s}^{-1}\,\mathrm{sr}^{-1}\,\mathrm{eV}^{-1}$. The integral energy flux remains within $1\text{–}6 \times 10^7\,\mathrm{keV}\,\mathrm{cm}^{-2}\,\mathrm{s}^{-1}\,\mathrm{sr}^{-1}$ during the broad-energy precipitation phase, but then drops between $10^6$ and $10^7\,\mathrm{keV}\,\mathrm{cm}^{-2}\,\mathrm{s}^{-1}\,\mathrm{sr}^{-1}$ during the phase when field lines passing by $S_1$ become dipolar ($t = $1800–1900 s). When precipitation observations from $S_1$ resume ($t \approx 1930\,\mathrm{s}$), the integral energy flux remains close to $10^7\,\mathrm{keV}\,\mathrm{cm}^{-2}\,\mathrm{s}^{-1}\,\mathrm{sr}^{-1}$ before briefly increasing to reach $10^8\,\mathrm{keV}\,\mathrm{cm}^{-2}\,\mathrm{s}^{-1}\,\mathrm{sr}^{-1}$ and abruptly decreasing around $10^6\,\mathrm{keV}\,\mathrm{cm}^{-2}\,\mathrm{s}^{-1}\,\mathrm{sr}^{-1}$ until the end of the simulation. Those integral energy flux values are in agreement with the Hardy model, according to which the nightside maximum total energy flux ranges between about $4 \times 10^7$ and $2 \times 10^8\,\mathrm{keV}\,\mathrm{cm}^{-2}\,\mathrm{s}^{-1}\,\mathrm{sr}^{-1}$, depending on the Kp index value (Hardy et al., 1989).

At virtual spacecraft $S_2$ (panels b and d), proton precipitation above 1 keV appears around $t = 1760\,\mathrm{s}$ in the simulation, corresponding roughly to the time when the broad-energy precipitation at $S_1$ becomes narrower in energy. For about one minute, broad-energy precipitation with up to 30 keV protons and with a mean energy of 6–7 keV is observed, leading to

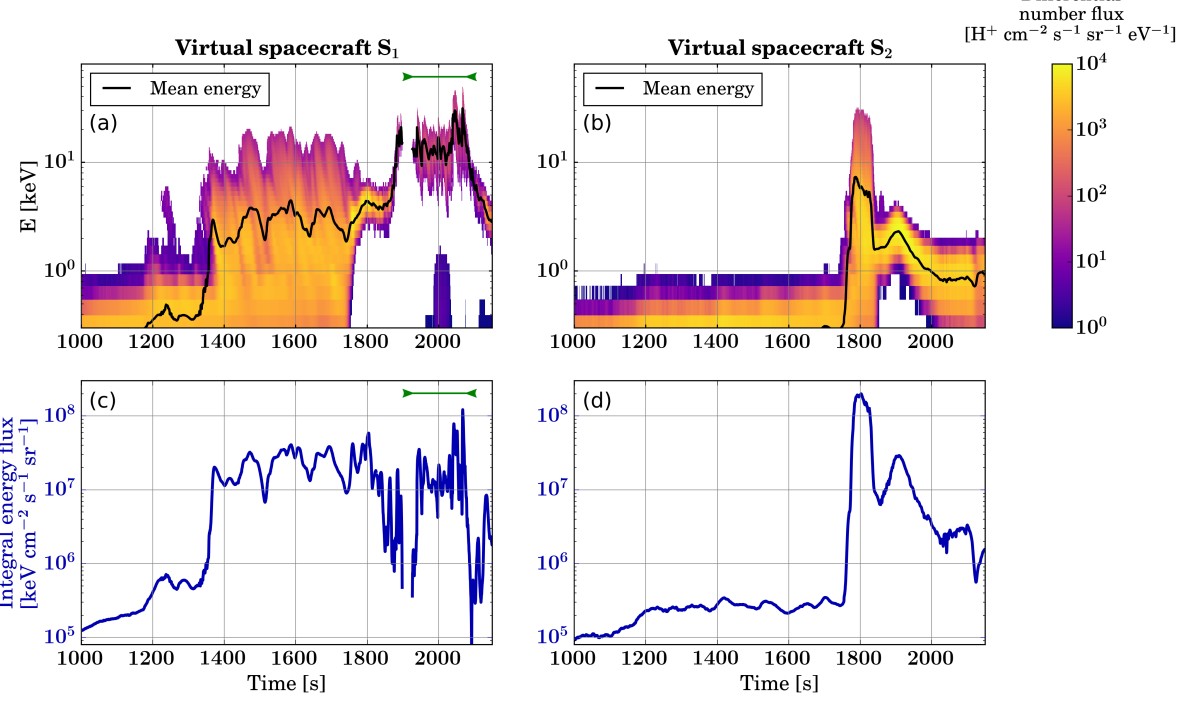

**Figure 5.** (a) Differential number flux of precipitating protons observed at virtual spacecraft $S_1$ as a function of time during the simulation. The black line indicates the mean precipitating energy as a function of time. (b) Same but at virtual spacecraft $S_2$. (c) Integral energy flux associated with proton precipitation as a function of time, at virtual spacecraft $S_1$. (d) Same but at virtual spacecraft $S_2$. The green segments in panels (a) and (c) indicate the time interval with precipitation associated with dipolarising flux bundles presented later.

integral flux values reaching $2 \times 10^8 \, \text{keV cm}^{-2} \, \text{s}^{-1} \, \text{sr}^{-1}$. The precipitating energy spectrum then becomes narrower and centred around 2 keV (during $t = 1850$–1900 s) and gradually broadens again until the end of the simulation, when the mean precipitating energy decreases below 1 keV.

While the 2D simulation setup with a scaled line dipole does not enable a direct mapping of the virtual spacecraft locations in terms of geomagnetic latitudes or even L values, it appears clearly that the properties and the dynamics of proton precipitation are very different at $S_1$ and $S_2$. From the geometry, $S_1$ maps to higher geomagnetic latitudes than $S_2$, and the analysis of Fig. 5 suggests that $S_1$ represents well geomagnetic latitudes usually mapping to the auroral oval, while $S_2$ would correspond to slightly lower latitudes where the proton auroral arc can be observed around the onset of ionospheric substorms, after drifting equatorwards during the growth phase (Liu et al., 2007). Indeed, precipitation is observed at $S_1$ before the global magnetotail reconfiguration due to reconnection is initiated (around $t = 1660$ s; Palmroth et al. (2017)), which suggests that $S_1$ maps to geomagnetic latitudes where aurora is visible during the growth phase of substorms. On the other hand, proton precip-

itation is only observed at $S_2$ around the time when the first particles accelerated earthwards following the global magnetotail reconfiguration reach the inner magnetosphere.

## 3.2 Flow bursts and precipitation

In what follows, we will focus on the proton precipitation associated with dipolarising flux bundles by considering virtual spacecraft $S_1$ between $t = 1920\,\mathrm{s}$ and $t = 2080\,\mathrm{s}$. During this time span, $S_1$ is on magnetic field lines mapping to the transition region near $X = -9R_E$, as can be seen in supplementary animation S2. Juusola et al. (2018a) showed that, during this time period, sustained tail reconnection is taking place at a dominant X-line drifting from $X \approx -14R_E$ to $X \approx -18R_E$ and is associated with fast earthward flows on the earthward side of the X-line.

Figure 6 is a snapshot of supplementary animation S2 at $t = 1945\,\mathrm{s}$. The colour-coded parameter is the $x$-component of the plasma bulk velocity, $V_x$. The location of virtual spacecraft $S_1$ is indicated with a red and white circle. Eight plus signs of various colours indicate locations of additional virtual spacecraft which will be used in the following to track the field-aligned component of the plasma bulk velocity. These virtual spacecraft are arranged in such a way that one can follow the bulk properties of the plasma between $S_1$ and the current sheet through the transition region. As can be seen in the figure, they are not located on a same field line, but rather along the region of positive $V_x$, which corresponds to the path followed by precipitating protons as field lines are being convected earthwards. Full velocity distributions are not available at those points due to limitations in the file sizes. The spacecraft indicated with an orange plus sign, which is the furthest in the current sheet, will be called $S_0$.

Figure 7 shows the precipitation flux observed at $S_1$ between $t = 1920\,\mathrm{s}$ and $t = 2080\,\mathrm{s}$, corresponding to the time interval marked with a green line in panels (a) and (c) of Fig. 5. Panel (a) of Fig. 7 shows the differential number flux of precipitating protons (colour scale) and the mean precipitating energy (black line). One can visually identify successive bursts of precipitating protons at energies ranging between about 5 keV and 50 keV. The energy dispersion of precipitating protons is visible, as within a given precipitation burst the highest energies are observed first and the lowest energies are observed last. This is also visible in the time variations of the mean precipitating energy. Between $t = 1985\,\mathrm{s}$ and $t = 2035\,\mathrm{s}$, a low-energy ($< 1\,\mathrm{keV}$) population of precipitating protons is observed in addition to the $\sim 10$ keV precipitation, but with fluxes lower than the main precipitating population around 10 keV by almost two orders of magnitude. The differential flux of the main precipitating proton population at 5–50 keV has values around $10^2\,\mathrm{cm^{-2}\,s^{-1}\,sr^{-1}\,eV^{-1}}$, which is relatively low compared to flux values at other times in the simulation that can reach $10^4\,\mathrm{cm^{-2}\,s^{-1}\,sr^{-1}\,eV^{-1}}$ (cf. Fig. 5a). Panel (b) shows the integral energy flux (blue line) alongside the component of the plasma bulk velocity at $S_1$ parallel to the magnetic field (red line), $V_\parallel$. Signatures of the precipitation bursts seen in panel (a) can be identified in both the integral energy flux and the plasma parallel bulk velocity. This suggests that the parallel bulk velocity can be used as a proxy for precipitating protons in this context.

In panel (c), the time series of the plasma parallel bulk velocity at the virtual spacecraft shown in Fig. 6 are indicated with a colour code consistent with that of the symbols indicating virtual spacecraft locations. The thick red line corresponds to the parallel velocity at $S_1$, i.e., shows the same data as the red line in panel (b). A given fluctuation in the parallel velocity at $S_1$ can be traced back in time and tailwards by visually identifying its corresponding signature at successive tailward virtual

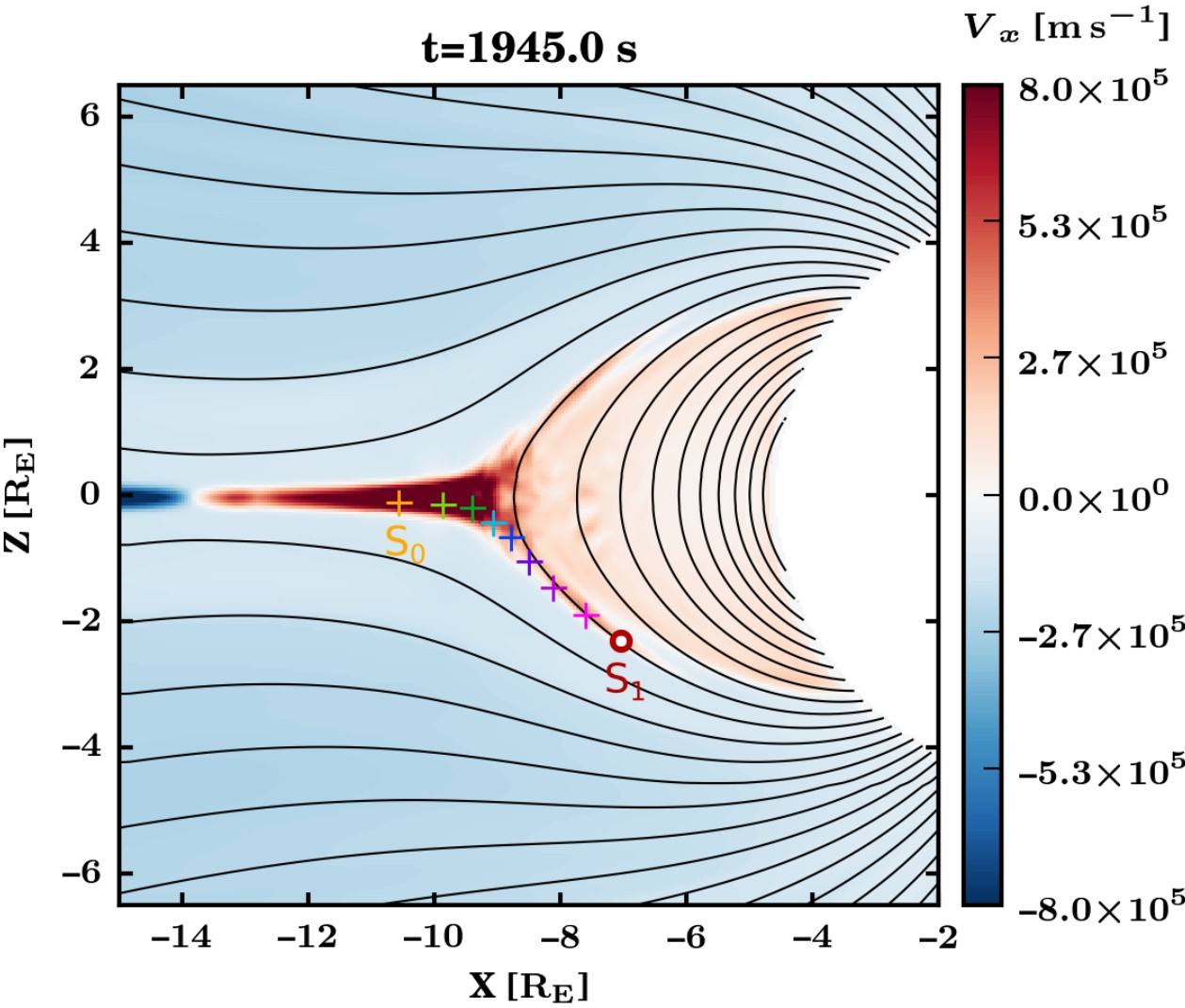

**Figure 6.** $x$-component of the plasma bulk velocity in the nightside part of the simulation plane at $t = 1945.0$ s. The red circle indicates the location of virtual spacecraft $S_1$, and the eight coloured plus signs show additional virtual spacecraft for which the parallel velocity components are shown in Fig. 7. Black lines represent magnetic field lines.

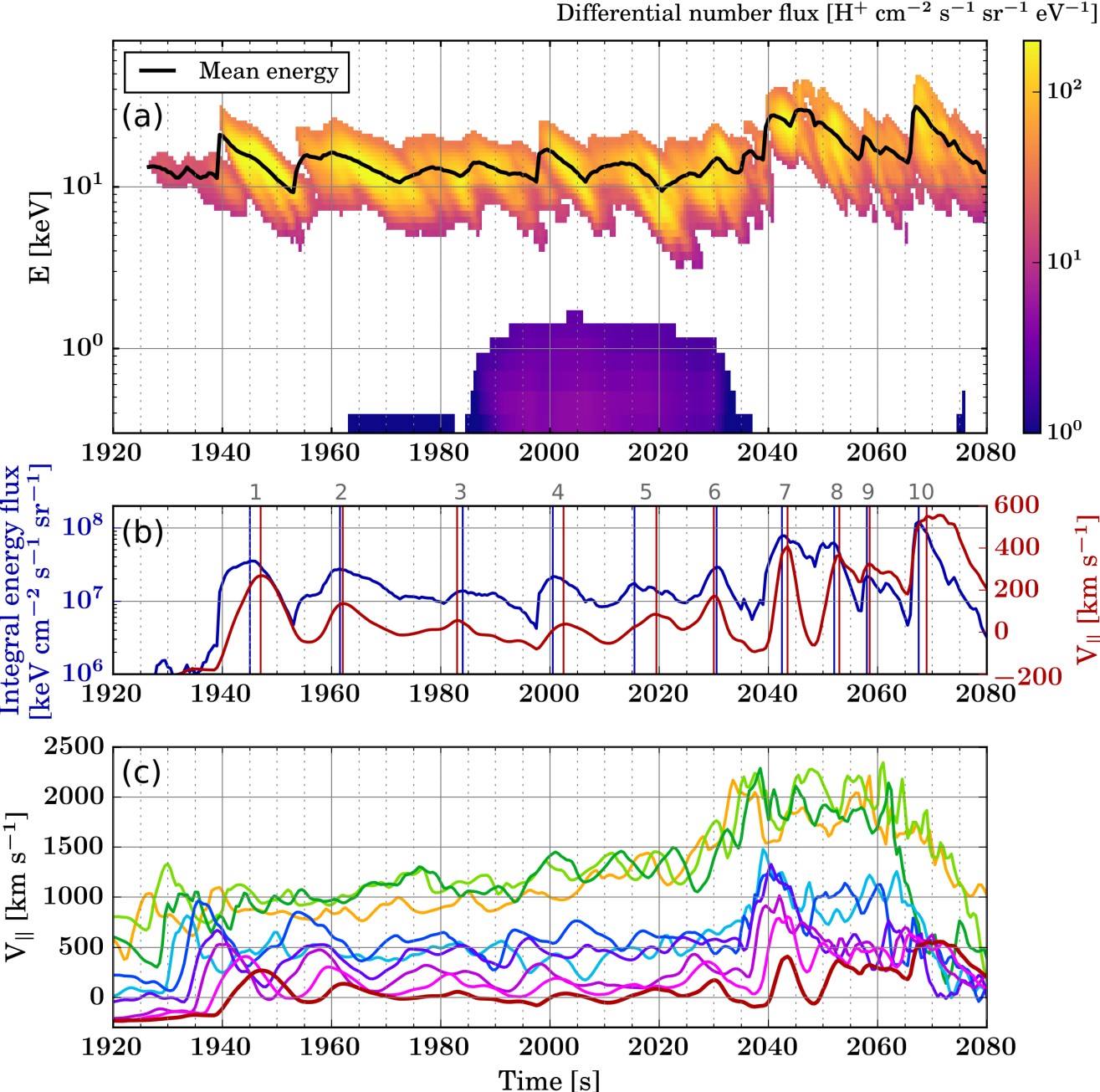

**Figure 7.** (a) Differential number flux of precipitating protons observed at virtual spacecraft $S_1$ between $t = 1920\,\text{s}$ and $t = 2080\,\text{s}$. The black line indicates the mean precipitating energy as a function of time. (b) Blue line: Integral energy flux associated with proton precipitation as a function of time at virtual spacecraft $S_1$. Red line: Parallel component of the plasma bulk velocity at $S_1$. The vertical lines indicate the times associated with peak values for these two parameters, and the numbers in grey above the panel identify the precipitation bursts discussed in the text. (c) Parallel component of the plasma bulk velocity at $S_1$ and at the virtual spacecraft indicated with plus signs in Fig. 6, with a consistent colour code.

**Table 1.** Times of peak integral energy flux and plasma bulk parallel velocities at $S_1$ and $S_0$ for the main bursts of proton precipitation. In the last column, X indicates whenever a given precipitation burst observed at $S_1$ could not be traced back to $S_0$.

| Burst number | Time of integral flux peak [s] | Time of $S_1$ $V_{\parallel}$ peak [s] | Time of $S_0$ $V_{\parallel}$ peak |
|---|---|---|---|
| 1 | 1945.0 | 1947.0 | 1926.5 |
| 2 | 1961.5 | 1962.0 | 1939.0 |
| 3 | 1984.0 | 1983.0 | X |
| 4 | 2000.5 | 2002.5 | 1985.0 |
| 5 | 2015.5 | 2019.5 | 1996.5 |
| 6 | 2030.5 | 2030.0 | X |
| 7 | 2042.5 | 2043.5 | 2033.5 |
| 8 | 2052.0 | 2053.0 | X |
| 9 | 2058.0 | 2058.5 | X |
| 10 | 2067.5 | 2069.0 | 2056.0 |

spacecraft. For instance, the parallel velocity enhancement peaking at $S_1$ around $t = 1947\,\mathrm{s}$ is consistent with parallel velocity enhancements visible at each of the virtual spacecraft, from the magenta one which is the closest to $S_1$ till the orange one ($S_0$) which is the deepest in the current sheet, peaking around $t = 1927\,\mathrm{s}$. This suggests that some dipolarising flux bundles, which are identified through short-lived enhancements in $V_x \approx V_{\parallel}$ in the current sheet, directly lead to proton precipitation into

the ionosphere in the regions mapped to the current sheet. However, not all the $V_{\parallel}$ enhancements at $S_1$ can be related to $V_x$ enhancements in the current sheet. For instance, the $V_{\parallel}$ peak at $S_1$ around $t = 2030\,\mathrm{s}$ cannot be traced back to a specific $V_{\parallel}$ peak at $S_0$.

Table 1 gives the peak times of the integral energy flux, plasma bulk parallel velocity at $S_1$ and plasma bulk parallel velocity at $S_0$ for the ten main precipitating flux enhancements between $t = 1920\,\mathrm{s}$ and $t = 2080\,\mathrm{s}$. These times correspond to those

of local maxima of each parameter, as indicated with vertical lines in panel (b) of Fig. 7 for the integral energy flux and the parallel velocity at $S_1$. For the four precipitation bursts observed at $S_1$ which could not be traced back to $S_0$ in the current sheet, the time of $S_0$ $V_{\parallel}$ peak is replaced with an X in the last column. In all cases but one, the peak times for the integral energy flux and $V_{\parallel}$ at $S_1$ are the same within 2 s; the exception is burst number 5 for which the triple peak in flux is associated with a single broad peak in $V_{\parallel}$, with 4 s difference in the times of local maxima.

In the four cases where a precipitation burst cannot be directly linked to a DFB passing by $S_0$, the signatures are lost between the cyan and blue spacecraft (burst number 3), between the blue and indigo spacecraft (burst number 6), between the indigo and purple spacecraft (burst number 8), and between the light and dark green spacecraft (burst number 9). These virtual spacecraft are located in the transition region, which suggests that the transition region can act like a buffer for DFBs, either directly transmitting them and leading to bursts of precipitating protons, or releasing precipitating particles with no direct correlation

with an incoming DFB from the tail.

## 4 Discussion

This paper presents the first unambiguous investigations of the global terrestrial magnetic field dynamics and its effects on proton precipitation in the midnight sector using a global hybrid-Vlasov model of the near-Earth environment, Vlasiator. Mende et al. (2002) listed four possible causes for particles to precipitate: (i) injection of particles on closed geomagnetic field lines, (ii) interaction of particles with electric fields, in particular waves, (iii) compression of the magnetic flux tube, and (iv) scattering on stretched magnetic field lines. All these causes can be studied using Vlasiator, except for some of the wave–particle interactions, as waves may not all be resolved by the model. It is known that proton precipitation can result from interactions with electromagnetic ion cyclotron (EMIC) waves (Erlandson and Ukhorskiy, 2001; Sakaguchi et al., 2008; Popova et al., 2018) as well as magnetosonic waves (Xiao et al., 2014). While magnetosonic waves are well resolved in Vlasiator, it is unclear whether this is also the case for EMIC waves, as their expected spatial scale is close to the ordinary-space grid resolution in the current runs. Wave-particle interactions are expected to be particularly important in the case of loss-cone scattering of trapped particles, i.e., ring current protons in the case of auroral proton precipitation. In this study, however, the analysis of Fig. 7 suggests that the precipitating protons essentially come from the magnetotail and are hence precipitating due to pitch-angle scattering resulting from the small curvature radius of the magnetic field near the neutral sheet (Sergeev and Tsyganenko, 1982). This is confirmed in Fig. S2 provided in Supplementary Material, which shows the value of the $\kappa$ parameter (see Introduction) along the $x$-axis between $-25R_E$ and $-6R_E$ as a function of time from $t = 1000$ s until the end of the simulation, as in Fig. 5. At the time of highest precipitating proton fluxes (i.e., after $t = 1800$ s) beyond $X \sim -10R_E$ in the plasma sheet, $\kappa$ exhibits mostly values below $\sqrt{8}$, hence fulfilling the Sergeev et al. (1983) criterion according to which protons get scattered into the loss cone on stretched field lines. While on the other hand it would prove interesting to estimate whether there is a non-negligible contribution to this precipitation from wave-particle interactions, this has to be left for a future study.

One limitation coming from the simulation set-up is related to the fact that the simulation run used in this study is 2D in ordinary space, which besides preventing a 3D description of the tail plasma requires using a line dipole instead of a point dipole for the geomagnetic field. The line dipole is scaled to reproduce a realistic dayside magnetopause stand-off distance, but it implies that geomagnetic field line equations have the form $r(\theta) = C \cos \theta$ in polar coordinates, with $C$ a constant and $\theta$ the latitude, instead of $r(\theta) = C \cos^2 \theta$ in the case of a centre dipole. This means that it is not possible to map L values with geomagnetic latitudes at ionospheric altitudes in this run; hence, we cannot assess how virtual spacecraft $S_1$ and $S_2$ are mapped to the ionosphere in terms of geomagnetic latitude. Such a discussion will be possible once 3D–3V Vlasiator runs are available. However, this 2D set-up still allows us to investigate the connection between tail dynamics, such as dipolarisation fronts, and precipitation.

Despite those limitations, the precipitating proton fluxes (differential number flux and integral energy flux) observed at the two selected locations (virtual spacecraft $S_1$ and $S_2$) are in agreement with predictions from the Hardy model (Hardy et al., 1989, 1991) in the midnight MLT sector and for active geomagnetic conditions (Kp > 3). While direct comparison with particle data from spacecraft would be desirable, the fact that DMSP satellites, which are currently the main source of direct

precipitating proton observations, are on Sun-synchronous orbits essentially near the dawn-dusk plane does not allow midnight-sector observations. In the nightside magnetosphere, comparison with data from the NASA Magnetospheric Multiscale (MMS) mission would prove interesting, but it is beyond the scope of the present study.

While velocity distributions are not saved in every simulation cell in Vlasiator runs, results suggest that the parallel component of the plasma bulk velocity can be used to at least qualitatively describe the integral energy flux of precipitating protons (cf. Fig. 7b). The idea that a beam superimposed to the core plasma population could lead to observable effects in the VDF first and second moments (bulk velocity and temperature) was brought forward by Parks et al. (2013), who showed that the apparent slowing down and temperature increase of the solar wind associated with nonlinear structures are due to the presence of a beam of particles propagating in opposite direction and with greater energy than the core solar wind population. In our case, precipitating proton beams along the magnetic field direction affect the parallel component of the plasma bulk velocity in an analogous manner. Future work could therefore include deriving a proxy for proton precipitation relying on plasma bulk parameters, which are saved in every simulation cells, to quantitatively estimate the precipitation parameters such as the integral energy flux or mean energy.

In their study of BBF-associated proton precipitation with a MHD model, Ge et al. (2012) found that dipolarisation fronts led to precipitating proton enhancements into the ionosphere with integral energy fluxes of the order of 0.1 $\mu$W m$^{-2}$. This corresponds to $6 \times 10^4$ keV cm$^{-2}$ s$^{-1}$. Assuming a uniform flux along all downwards directions as a rough approximation ($2\pi$ solid angle), this gives a directional integral energy flux of the order of $10^4$ keV cm$^{-2}$ s$^{-1}$ sr$^{-1}$, which is roughly three orders of magnitude below values obtained with Vlasiator when the global magnetotail reconfiguration is taking place (cf. Fig. 7b). A likely explanation is that the MHD simulation has low ion temperature $T_i$ values compared to our kinetic approach, leading to lower precipitation energy fluxes as these are proportional to $T_i\sqrt{T_i}$ in their approach (cf. eq. (1) in Ge et al., 2012). We note that the Vlasiator integral flux values are on the other hand in agreement with the test-particle simulation results presented in their companion paper (Zhou et al., 2012b) with integral energy fluxes of 0.2–1 mW m$^{-2}$ (equivalent to $2 \times 10^7 - 10^8$ keV cm$^{-2}$ s$^{-1}$ sr$^{-1}$ with the same reasoning as above), and with statistical patterns shown in Galand et al. (2001) ($\sim$0.1 mW m$^{-2}$). One strength of evaluating the proton precipitation parameters using a kinetic model is that not only integral energy fluxes can be calculated, but also differential fluxes, which may enable more detailed future studies of proton precipitation and its link to global magnetospheric dynamics. Further, a test-particle approach is not fully self-consistent, in the sense that the electromagnetic fields affect the particle distributions but not vice versa. Therefore the test-particle approach does not fully describe, e.g., dynamics of reconnection-related precipitation.

The examination of precipitation bursts passing by virtual spacecraft S$_1$ between $t = 1920$ s and $t = 2080$ s suggests that these bursts are associated with dipolarising flux bundles originating from the vicinity of the stable X-line in the current sheet. Field-aligned beams of plasma propagating earthwards associated to dipolarisation fronts have been observed and simulated not only in the central plasma sheet, but also in the plasma sheet boundary layer (PSBL) (Zhou et al., 2012a). In this Vlasiator run, there is unfortunately no cell located in the PSBL where the full VDF is saved at each time step to make a comparison with results from Zhou et al. (2012a), hence we focus on proton precipitation originating from the central plasma sheet. Dipolarisation front signatures in the $B_z$ component of the current-sheet plasma can be identified in Fig. 6 of Juusola et al.

(2018a) during the studied time interval (written as $t = 32{:}00$–$34{:}40$ in their figure). While there are not enough precipitation bursts at $S_1$ in the studied time period to carry out a statistical analysis of their properties (integral flux enhancement and parallel velocity enhancement at $S_1$, $v_x$ enhancement at $S_0$), it can be noted that there does not seem to be a one-to-one correlation between the magnitude of the $v_\parallel$ enhancement at $S_1$ and the possibility to trace it back to $S_0$. This suggests that the transition region, corresponding to the location in the magnetotail where tail-like geomagnetic field lines become more dipolar (near $X = -10.5\,R_E$ in Fig. 3), plays a role in regulating these bursts. It is known that fast flows associated with BBFs can bounce when reaching the inner magnetosphere (e.g., Ohtani et al., 2009; Juusola et al., 2013; Nakamura et al., 2013) and may even exhibit multiple overshoots and oscillations around their equilibrium position (Panov et al., 2010). Our results further indicate that contrary to a frequent assumption, the transition region may be more than just a passive mediator between the plasma sheet and the ionosphere. DFBs could perhaps themselves experience some form of bouncing near the transition region, which could explain the regulation of the studied proton precipitation bursts.

Auroral activations concurrent with BBFs have been widely studied (e.g., Nakamura et al., 2001a; Sergeev et al., 2004; Gallardo-Lacourt et al., 2014). However, the time scales that are involved in such processes are of the order of up to 10 min, in contrast to the short-lived DFBs and associated proton precipitation bursts ($< 1\,\mathrm{min}$ duration) studied here. In terms of auroral emissions observed from the ground, it must be noted that the precipitation bursts observed at $S_1$ might actually lead to more weakly modulated emissions than can be inferred from the integral energy flux variations at the virtual spacecraft, as the energy dispersion of precipitating protons tends to smooth the integral energy flux as particles get closer to Earth. This warrants future studies involving ground-based optical observations of proton aurora at high enough cadence (a few seconds at most) to investigate whether DFB-related proton aurora signatures can be seen from the ground. If not, this would imply that global ion-kinetic magnetospheric simulations, supplemented by data from spacecraft orbiting in the magnetosphere, might be the only tool to investigate the active role played by the transition region in regulating the precipitation of auroral protons to the nightside ionosphere.

## 5 Conclusions

This paper presents the first evaluations of auroral ($\sim$1–30 keV) proton precipitation from a global hybrid-Vlasov magnetospheric model, Vlasiator. The simulation run considered here corresponds to relatively fast solar wind ($750\,\mathrm{km\,s^{-1}}$) with purely southward IMF of moderate magnitude ($|\mathbf{B}| = |B_z| = 5\,\mathrm{nT}$).

The evaluation of the differential number flux of precipitating protons at a given location in the nightside magnetosphere is achieved by averaging the velocity distribution function inside the bounce loss cone within energy bins. The integral energy flux of precipitation as well as the mean precipitating energy are also calculated from the differential number flux. The main results of this study can be summarised as follows.

1. From this first case study, we find that Vlasiator reproduces auroral proton precipitation with realistic differential number fluxes and energies, when compared to the Hardy model.

2. During a selected time interval when a single X-line dominates the tail reconnection in the simulation, proton precipitation observed at a virtual spacecraft in the inner magnetosphere occurs in a bursty manner. In this situation, the integral precipitating energy flux exhibits variations mostly similar to the parallel component of the plasma bulk velocity at the virtual spacecraft. This suggests that the integral energy flux can qualitatively be described with the local parallel velocity at locations for which full velocity distributions are not saved in the Vlasiator run.

3. Finally, it is found that proton precipitation bursts can in some cases be traced back to the current sheet and are associated with dipolarising flux bundles. However, not all precipitation bursts correspond to a definite DFB, which suggests that the transition region plays a role in regulating auroral proton precipitation associated with DFBs during BBFs.

*Code availability.* Vlasiator is an open source code released under the GPLv2 license. The code is available at http://github.com/fmihpc/vlasiator.

*Author contributions.* MG carried out most of the analysis and prepared the manuscript. YP-K participated in running the simulation and development of the analysis methods. UG and MB participated in the development of the analysis methods. All co-authors helped in the interpretation of the results, read the manuscript and commented on it.

*Competing interests.* The authors declare that they have no conflict of interest.

*Acknowledgements.* We acknowledge The European Research Council for Starting grant 200141-QuESpace, with which the Vlasiator model (http://www.physics.helsinki.fi/vlasiator) was developed, and Consolidator grant 682068-PRESTISSIMO awarded for further development of Vlasiator and its use in scientific investigations. We gratefully acknowledge Academy of Finland grants number 267144, 312351, and 309937. PRACE (http://www.prace-ri.eu) is acknowledged for granting us Tier-0 computing time in HLRS Stuttgart, where Vlasiator was run in the HazelHen machine with project number 2014112573. The work of LT is supported by a Marie Sklodowska-Curie Individual Fellowship (#704681). M. G. thanks Prof. Rami Vainio for useful discussions on the mathematical formulation of the directional differential flux estimation.

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
