# Peer review of "Hybrid-Vlasov modelling of nightside auroral proton precipitation during southward interplanetary magnetic field conditions"

_Annales Geophysicae, 2019_

## Referee Comment (RC1) · Anonymous Referee #1 · 28 May 2019

This is interesting and important study of ion precipitations in the magnetotail. Authors use quite developed simulation tool to check the effect of fast plasma flows on formation of precipitating ion fluxes. The brief comparison of simulation results and published observational data shows a reasonable agreement. I believe this paper should be published in Angeo after Authors address several (quite minor) questions.

Introduction: there are several important references related to proton aurora investigations that may be included: 10.1029/2008JA013099, 10.1134/S001679321805016X, 10.1134/S0016793218040114

Page 7, eq. (8): this equations assume that ion energy is conversed along the bounce

trajectory, i.e. there is no field-aligned electric fields in the system. It would be useful to show 2D plot with the parallel electric field distribution and quickly discuss the weakness of this electric field effect on ion dynamics.

Page 8, Line 17: <1o of the loss-cone is an estimate based on nondisturbed magnetic field models. It would be useful to provide also a loss-cone estimate for magnetic field enhancements at the dipolarization front accompanied fast flows (where Bz is significantly larger than the magnetotial Bz)

Page 17, Lines 1-5: Previous models of pitch-angle scattering (e.g., Sergeev and Tsyganenko, 1982) were developed for the quiet time magnetotail current sheet, whereas in this paper Authors consider ion precipitation from the acceleration (fast plasma flow) region, what is closer to simulation results shown in 10.1029/2012JA018171, 10.1029/2012JA017677. Any relations to pitch-angle scattering on the magnetic field line curvature should be confirmed by corresponding estimates of the kappa parameter, e.g. kappa dependence on x and time would support Authors' conclusions.

There are two model features that require some explanations/discussions: Figure 3: what is a local temperature minimum around S1 position? Temperatures earthward and tailward from S1 are higher than in the S1 location. Is there any analogy of such temperature minimum in the statistical spacecraft observations? As I know, the temperature profile along the magnetotail is generally monotonous (see, e.g., 10.1029/2008JA013849, 10.1002/2016JA023710.)

Figure 4: some of shown distributions are definitely unstable (they contain ion beams with positive slobs along the parallel velocity direction). Thus, some discussion is needed to explain if these instabilities are too slow to influence ion distributions or they are simply suppressed in the numerical calculations.

---

## Referee Comment (RC2) · Anonymous Referee #2 · 11 Jun 2019

The manuscript deals with the nightside proton precipitation, focusing on the perspective of the magnetosphere, by using Vlasiator, a global kinetic hybrid simulation of the near-Earth environment. The authors found a good agreement in terms of differential number fluxes and energies, compared to the empirical Hardy model. The proton precipitations are observed as burst events and, in some cases, they can be traced back to the current sheet in the magnetotail and associated with depolarising flux bundles.

The paper brings interesting results and can be acceptable for publication after some clarifications.

1. In my opinion, the authors should justify the resolutions chosen for the simulation

in terms of characteristic plasma quantities. In particular, in order to comment on the physical space resolution, what is the value for the ion inertial length? or Larmor radius? Since the proton precipitation is produced by the magnetic reconnection in the magnetotail, how thin is the current sheet? And, in a similar way, the authors should comment on the velocity space resolution. In this respect, what is the value for the Alfvén speed?

2. In the inner boundary, protons are described with a static Maxwellian VDF. This means that here kinetic effects are neglected. However, the observed VDFs close to the inner boundary, for example in S2 (Fig.3), are strongly non Maxwellian during the precipitation. I am wondering if the imposed sharp change in the VDF could influence the results. Could the authors comment on this point?

3. One of the hypotheses used to evaluate the directional differential particle flux is that 'protons remain attached to a given magnetic flux tube'. However, during a magnetic reconnection event this is not exactly true. Although the locations in the magnetosphere are chosen far from the X-point, I am wondering if the change in the magnetic topology can have effects also at these points for the analysis.

4. During the phase 2, the orbit of the NASA-MMS mission was chosen to spend time on the night side of the Earth's magnetosphere. Did the authors check if there are any datasets able to support their results?

---

## Author Comment (AC1) · 28 Jun 2019

We thank the Reviewer for their careful analysis of our manuscript as well as their valuable comments and suggestions. Our preliminary responses to the comments are shown in blue, while the original text written by the Reviewer is shown in black.

This is interesting and important study of ion precipitations in the magnetotail. Authors use quite developed simulation tool to check the effect of fast plasma flows on formation of precipitating ion fluxes. The brief comparison of simulation results and published observational data shows a reasonable agreement. I believe this paper should be published in Angeo after Authors address several (quite minor) questions.

[Figure]

Introduction: there are several important references related to proton aurora investigations that may be included: 10.1029/2008JA013099, 10.1134/S001679321805016X, 10.1134/S0016793218040114.

Thank you for suggesting those references, which can indeed be added in the introduction.

Page 7, eq. (8): this equation assumes that ion energy is conserved along the bounce trajectory, i.e. there is no field-aligned electric fields in the system. It would be useful to show 2D plot with the parallel electric field distribution and quickly discuss the weakness of this electric field effect on ion dynamics.

We can add to Supplementary Material Figure 1 below showing the parallel component of the electric field at the same time step and with a similar format as Fig. 3. This parallel component was averaged over 120 s, which corresponds roughly to a quarter of the bounce period for 10 keV protons at $L = 9$ (virtual spacecraft S1).

As can be seen in the figure, the parallel electric field between S1 or S2 and the inner boundary is of the order of 1e-5 to 1e-4 V/m. When integrated along the field line between S1 and the inner boundary, this corresponds to a potential difference of the order of 1 kV. In their discussion of the effect of potential drops in the auroral acceleration region on precipitating protons, Liang et al. (2013, 10.1002/jgra.50454) estimate that for ions with energies $\gg 1$ keV the acceleration resulting from typical potential drops in the AAR ($\sim$1 kV up to $\sim$4 kV occasionally) can be neglected, which enables a reasonable mapping of auroral latitudes to the central plasma sheet.

The above discussion can be added in a revised version of the manuscript.

Page 8, Line 17: <1° of the loss-cone is an estimate based on nondisturbed magnetic field models. It would be useful to provide also a loss-cone estimate for magnetic field enhancements at the dipolarization front accompanied fast flows (where Bz is significantly larger than the magnetotail Bz).

This is a good point. Using eq. (5), we can estimate the value of the loss-cone angle at a dipolarisation front in the equatorial plane whose Bz component is, e.g., 30 nT (see, e.g., Juusola et al., 2018a, 10.5194/angeo-36-1183-2018, or Runov et al., 2017, 10.1002/2017JA024010). Assuming a mapping to auroral latitudes ($B_0$ of the order of 6e4 nT), we obtain a loss cone angle of about 1.3°. We can mention this point in a revised version of the paper.

Page 17, Lines 1-5: Previous models of pitch-angle scattering (e.g., Sergeev and Tsyganenko, 1982) were developed for the quiet time magnetotail current sheet, whereas in this paper Authors consider ion precipitation from the acceleration (fast plasma flow) region, what is closer to simulation results shown in 10.1029/2012JA018171, 10.1029/2012JA017677. Any relations to pitch-angle scattering on the magnetic field line curvature should be confirmed by corresponding estimates of the kappa parameter, e.g. kappa dependence on x and time would support Authors' conclusions.

Thank you for this suggestion. Figure 2 below shows the value of the kappa parameter along the x-axis between $-25R_E$ and $-6R_E$ (which corresponds to the regions of interest in our study) as a function of time from $t = 1000$ s until the end of the simulation, as in Fig. 5.

As can be seen in this figure, at the time of highest precipitating proton fluxes (i.e., after $t = 1800$ s) beyond $X \sim -10R_E$ in the plasma sheet, kappa exhibits mostly values below $\sqrt{8}$, hence fulfilling the Sergeev et al. (1983) criterion according to which protons get scattered into the loss cone on stretched field lines. This indeed confirms that this mechanism is the cause for auroral proton precipitation in this Vlasiator simulation.

The two references suggested by the Reviewer can be discussed in a revised version of the manuscript, and the figure showing kappa could be added as supplementary material.

There are two model features that require some explanations/discussions: Figure 3: what is a local temperature minimum around S1 position? Temperatures earthward

and tailward from S1 are higher than in the S1 location. Is there any analogy of such temperature minimum in the statistical spacecraft observations? As I know, the temperature profile along the magnetotail is generally monotonous (see, e.g., 10.1029/2008JA013849, 10.1002/2016JA023710.)

First of all, something to note is that the shown temperature is the temperature of an isotropic plasma which would have the same total pressure as the simulated distribution, and thus does not represent the temperature of the bulk plasma but rather the measured effect of a combined bulk plasma and fast additional flow.

The local temperature minimum seen in Fig. 3 comes from the fact that hot plasma associated with the precipitating protons originates from the current sheet region, which leads to temperature enhancements propagating earthwards in terms of L shells. At the time step chosen for Fig. 3 (1800 s), a stream of hot ($T = 2\mathrm{e}7$ K) plasma coming from the transition region is reaching S1, leading to a local enhancement of the proton temperature compared to its background value at S1.

It is difficult to compare our results to those shown in the two suggested references, as in 10.1029/2008JA013849 the statistical observations are obtained with slow flows ($V_\perp < 150$ km/s) whereas in the situation shown in our Fig. 3 $V_\perp$ is essentially greater than 200 km/s for $X < -7R_E$. As for the $T_i$ profile along the magnetotail shown in 10.1002/2016JA023710, it consists of only three points with $X$ between $-10R_E$ and $-30R_E$, meaning that (i) the spatial resolution is too coarse to capture such small-scale variations as that pointed in our Fig. 3 and (ii) measurements at $X > -10R_E$ are not available.

Figure 4: some of shown distributions are definitely unstable (they contain ion beams with positive slobs along the parallel velocity direction). Thus, some discussion is needed to explain if these instabilities are too slow to influence ion distributions or they are simply suppressed in the numerical calculations.

Thank you for raising this issue. In Vlasiator, instabilities are resolved in the numerical

calculations provided their wavelengths are larger than the grid resolution in ordinary space. According to Gary (1989, 10.1007/BF00196632; see especially Table II p. 385), there are several instabilities which can arise from interactions between a proton beam and a proton core population in a plasma.

The ion/ion right-hand resonant instability ("magnetosonic" or "fast MHD"), of much lower frequency than the ion gyrofrequency, can develop when the field-aligned drift velocity $v_0$ of the protons is greater than the Alfvén speed $v_A$, and when the ratio between beam ($n_b$) and core ($n_c$) densities is "very small". If we consider the distribution shown in Fig. 4a of our manuscript, we have $v_0 \sim 1000$ km/s, $v_A = 4379$ km/s, $n_b = 1.1\mathrm{e}4$ m$^{-3}$ and $n_c = 1.2\mathrm{e}5$ m$^{-3}$, i.e., $v_0/v_A \sim 0.2$ and $n_b/n_c = 8.6\mathrm{e}\text{-}2$. Since the condition on the velocities is not fulfilled, it is unlikely that this instability grows at the location where the VDF is observed.

The ion/ion left-hand instability also requires $v_A < v_0$, and in addition it requires that the ion beam to be hot, i.e., with a thermal velocity greater than the beam drift velocity. This instability is therefore not expected to grow in the considered situation of Fig. 4a.

The ion/ion nonresonant instability ("firehose") can develop if $v_A \ll v_0$. Hence, it cannot grow in this situation either.

In conclusion, in the $v_A > v_0$ regime which characterises the plasma in the vicinity of virtual spacecraft S1 and S2, we do not expect that the instabilities listed above can grow fast enough to significantly affect the ion distributions on the time scale needed for precipitating protons (i.e., essentially the field-aligned beam) to reach the inner boundary or even the ionosphere.

We can include a shorter version of this discussion in a revised version of the manuscript.
* * *
[Figure]

**t=1800.0 s**

**Epara [V/m]**

Parallel electric field plot with axes X [R_E] from -14 to -2 and Z [R_E] from -6 to 6.

Colorbar values:
$1.0 \times 10^{-2}$
$1.0 \times 10^{-3}$
$1.0 \times 10^{-4}$
$1.0 \times 10^{-5}$
$0.0 \times 10^{0}$
$-1.0 \times 10^{-5}$
$-1.0 \times 10^{-4}$
$-1.0 \times 10^{-3}$
$-1.0 \times 10^{-2}$

**Fig. 1.** Parallel electric field at t = 1800 s, averaged over 120 s

[Figure]

**Fig. 2.** Keogram of the kappa parameter along the nightside x-axis

---

## Author Comment (AC2) · 28 Jun 2019

We thank the Reviewer for their careful analysis of our manuscript as well as their constructive comments and suggestions. We provide our preliminary responses to the comments in blue, while the original text written by the Reviewer is shown in black.

The manuscript deals with the nightside proton precipitation, focusing on the perspective of the magnetosphere, by using Vlasiator, a global kinetic hybrid simulation of the near-Earth environment. The authors found a good agreement in terms of differential number fluxes and energies, compared to the empirical Hardy model. The proton precipitations are observed as burst events and, in some cases, they can be traced back

to the current sheet in the magnetotail and associated with depolarising flux bundles.

The paper brings interesting results and can be acceptable for publication after some clarifications.

1. In my opinion, the authors should justify the resolutions chosen for the simulation in terms of characteristic plasma quantities. In particular, in order to comment on the physical space resolution, what is the value for the ion inertial length? or Larmor radius? Since the proton precipitation is produced by the magnetic reconnection in the magnetotail, how thin is the current sheet? And, in a similar way, the authors should comment on the velocity space resolution. In this respect, what is the value for the Alfvén speed?

Thank you for this nice suggestion. We are indeed happy to provide these values and put them in perspective with the ordinary and velocity space resolutions. We can for instance provide such values in the solar wind (input) and in the transition region (at $X = -10R_E$, $Y = Z = 0$) at $t = 1800$ s (Fig. 2).

– Solar wind:
Ion inertial length $\lambda_d = 228$ km
Ion Larmor radius $r_L = 214$ km
Alfvén speed $v_A = 109$ km/s

– Transition region:
Ion inertial length $\lambda_d = 906$ km
Ion Larmor radius $r_L = 681$ km
Alfvén speed $v_A = 1596$ km/s

The real space resolution of 300 km in this run is therefore sufficient to resolve most of the proton kinetics, as was confirmed in a dedicated study by Pfau-Kempf et al. (2018, 10.3389/fphy.2018.00044).

The current sheet thickness was evaluated during this same run by Juusola et al.

[Figure]

(2018b, 10.5194/angeo-36-1027-2018) and is shown in Fig. 6 of their paper. In the region of interest for our study ($X > -20R_E$), Juusola et al. find that the plasma sheet thickness essentially lies within 0.1–0.2 $R_E$ (i.e., just a few simulation cells).

The above values can be provided and discussed in a revised version of the manuscript.

2. In the inner boundary, protons are described with a static Maxwellian VDF. This means that here kinetic effects are neglected. However, the observed VDFs close to the inner boundary, for example in S2 (Fig.3), are strongly non Maxwellian during the precipitation. I am wondering if the imposed sharp change in the VDF could influence the results. Could the authors comment on this point?

The imposed static Maxwellian VDF at the inner boundary mostly affects the neighbouring cells through the calculation of translation and acceleration terms in the Vlasov equation. Since S2 is located about 15 cells away from the inner boundary, its stencil used for acceleration and translation do not reach the inner boundary. Given that, in addition, the plasma flows from S2 towards the inner boundary, potential diffusion effects are likely negligible in comparison with the bulk flow.

3. One of the hypotheses used to evaluate the directional differential particle flux is that 'protons remain attached to a given magnetic flux tube'. However, during a magnetic reconnection event this is not exactly true. Although the locations in the magnetosphere are chosen far from the X-point, I am wondering if the change in the magnetic topology can have effects also at these points for the analysis.

Since the locations of virtual spacecraft S1 and S2 are on closed field lines which do not further reconnect and are convected earthwards, there is no risk that the precipitating protons observed at S1 (or S2) are affected by magnetic reconnection before they reach the ionosphere. Hence, there should be no concerns regarding the validity of the hypothesis that the plasma observed at the virtual spacecraft remains attached to its magnetic flux tube until precipitating protons reach the ionosphere.

4. During the phase 2, the orbit of the NASA-MMS mission was chosen to spend time on the night side of the Earth's magnetosphere. Did the authors check if there are any datasets able to support their results?

Comparing our results with MMS data would indeed be interesting; however, we feel that this task would be beyond the scope of the present study, whose main aim is twofold: (i) present the methodology to evaluate proton precipitation in Vlasiator simulations, and (ii) discuss the nightside proton precipitation from a global perspective during a simulated event with southward IMF. A comparison of those results with observations could therefore be carried out in a follow-up study.

---

## Editor Comment (EC1) · Anna Milillo (Editor) · 24 Jul 2019

Dear authors, given your answers to the referees' comments, I would ask you to provide the revised manuscript with highlighted changes, so that I can ask to the referees if they are satisfied with the new version Anna Milillo
* * *

---

## Author Response (AR1)

**Manuscript angeo-2019-59: "Hybrid-Vlasov modelling of nightside auroral proton precipitation during southward interplanetary magnetic field conditions" by M. Grandin et al.**

**Response to Reviewer #1**

We thank the Reviewer for their careful analysis of our manuscript as well as their valuable comments and suggestions. Our responses to the comments are shown in blue, while the original text written by the Reviewer is shown in black.

This is interesting and important study of ion precipitations in the magnetotail. Authors use quite developed simulation tool to check the effect of fast plasma flows on formation of precipitating ion fluxes. The brief comparison of simulation results and published observational data shows a reasonable agreement. I believe this paper should be published in Angeo after Authors address several (quite minor) questions.

Introduction: there are several important references related to proton aurora investigations that may be included: 10.1029/2008JA013099, 10.1134/S001679321805016X, 10.1134/S0016793218040114
Thank you for suggesting those references, which have been added in the introduction.
– p. 2, l. 20: "Electromagnetic ion cyclotron (EMIC) waves are the prime candidate for such interactions (e.g., Erlandson and Ukhorskiy, 2001; Sakaguchi et al., 2008; **Popova et al., 2018**), but more recently, Xiao et al. (2014) suggested that fast magnetosonic waves, also known as equatorial noise, might interact with protons across a broad range of magnetic local times."
– p. 2, l. 25: "**On the dayside, it has been found that proton aurora flashes can be observed equatorwards from the cusp in relation with EMIC waves associated with plasma pressure pulses in the magnetosphere (Yahnina et al., 2008). More generally, EMIC waves can be responsible for auroral proton precipitation equatorwards of the proton oval forming long-lasting spots, arcs and flashes (Yahnin et al., 2018).**"
The Popova et al. (2018) reference was also added on p. 19, l. 11.

Page 7, eq. (8): this equation assumes that ion energy is conserved along the bounce trajectory, i.e. there is no field-aligned electric fields in the system. It would be useful to show 2D plot with the parallel electric field distribution and quickly discuss the weakness of this electric field effect on ion dynamics.
We have added Fig. S1 below to Supplementary Material, showing the parallel component of the electric field at the same time step and with a similar format as Fig. 3. This parallel component was averaged over 120 s, which corresponds roughly to one bounce period for 10 keV protons at $L = 9$ (virtual spacecraft $S_1$).

We have added the following discussion of this supplementary figure to the manuscript:

– p. 12, l. 29: "**One assumption which is made during the derivation of the precipitating fluxes is that there are no field-aligned electric fields in the system (cf. eq. (8)). Figure S1 in the Supplementary Material shows the parallel component of the electric field at the same time step and with a similar format as Fig. 3. This parallel component was averaged over 120 s, which corresponds roughly to one bounce period for 10 keV protons at $L = 9$ (virtual spacecraft $S_1$). It can be seen that the parallel electric field between $S_1$ or $S_2$ and the inner boundary is of the order of 0.01 to 0.1 mV m$^{-1}$. When integrated along the field line between $S_1$ and the inner boundary, this corresponds to a potential difference of the order of 1 kV. In their discussion of the effect of potential drops in the auroral acceleration region on precipitating protons, Liang et al. (2013) estimate that for ions with energies >> 1 keV the acceleration resulting from typical potential drops in the auroral acceleration region (~1 kV up to ~4 kV occasionally) can be neglected, which enables a reasonable mapping of auroral latitudes to the central plasma sheet. It therefore seems acceptable to neglect the effect of parallel electric fields when deriving the precipitating proton fluxes with the method described in section 2.2.**"

[Figure]

**Figure S1.** Simulated parallel component of the electric field in the nightside magnetosphere at $t = 1800$ s and averaged over 120 s. The Sun is located beyond the right boundary of the figure. The two black circles filled with white are virtual spacecraft $S_1$ and $S_2$ . $R_E = 6371$ km.

Page 8, Line 17: <1° of the loss-cone is an estimate based on nondisturbed magnetic field models. It would be useful to provide also a loss-cone estimate for magnetic field enhancements at the dipolarization front accompanied fast flows (where Bz is significantly larger than the magnetotail Bz)

This is a good point. We have added a short discussion as suggested in the revised manuscript:

– p. 9, l. 1: "In the **undisturbed** nightside equatorial plane, $\theta_{lc}$ takes values of a few degrees at most, and is smaller than 1° beyond 10 $R_E$ (see, e.g., Sergeev and Tsyganenko, 1982). **In presence of a rather large dipolarisation front with, e.g., $B_0 \approx B_z = 30$ nT, assuming a mapping to auroral latitudes ($B_{sc} \sim 53\,000$ nT), eq. (11) gives $\theta_{lc} = 1.4°$.**"

Page 17, Lines 1-5: Previous models of pitch-angle scattering (e.g., Sergeev and Tsyganenko, 1982) were developed for the quiet time magnetotail current sheet, whereas in this paper Authors consider ion precipitation from the acceleration (fast plasma flow) region, what is closer to simulation results shown in 10.1029/2012JA018171, 10.1029/2012JA017677. Any relations to pitch-angle scattering on the magnetic field line curvature should be confirmed by corresponding estimates of the kappa parameter, e.g. kappa dependence on x and time would support Authors' conclusions.

Thank you for this suggestion. Below is a figure showing the value of the $\kappa$ parameter along the x-axis between $-25\ R_E$ and $-6\ R_E$ (which corresponds to the regions of interest in our study) as a function of time from $t = 1000$ s until the end of the simulation, as in Fig. 5.

[Figure]

**Figure S2.** $\kappa$ parameter along the *x*-axis between $-25\ R_E$ and $-6\ R_E$ as a function of time from $t = 1000$ s until the end of the simulation. The black contour shows the critical value $\sqrt{8}$ below which the criterion for particle pitch-angle scattering is verified (orange and red area).
Note: $\kappa = \sqrt{(R_c/r_L)}$, with $R_c$ the curvature radius of the magnetic field and $r_L$ the proton gyroradius (Sergeev et al., 1983).

We have added this figure in the Supplementary Material as Fig. S2, and we have added the following sentences to the Discussion:
– p. 19, l. 14: "In this study, however, the analysis of Fig. 7 suggests that the precipitating protons essentially come from the magnetotail and are hence precipitating due to pitch-angle scattering resulting from the small curvature radius of the magnetic field near the neutral sheet (Sergeev and Tsyganenko, 1982). **This is confirmed in Fig. S2 provided in Supplementary Material, which shows the value of the $\kappa$ parameter (see Introduction) along the *x*-axis between $-25\ R_E$ and $-6\ R_E$ as a function of time from $t = 1000$ s until the end of the simulation, as in Fig. 5. At the time of highest precipitating proton fluxes (i.e., after $t = 1800$ s) beyond $X \sim -10\ R_E$ in the plasma sheet, $\kappa$ exhibits mostly values below $\sqrt{8}$, hence fulfilling the Sergeev et al. (1983) criterion according to which protons get scattered into the loss cone on stretched field lines.** While **on the other hand** it would prove interesting to estimate whether there is a non-negligible contribution to this precipitation from wave-particle interactions, this has to be left for a future study."

The two references suggested by the Reviewer have been discussed in the revised manuscript as follows:

– p. 20, l. 18: "In their study of BBF-associated proton precipitation with a MHD model, Ge et al. (2012) found that dipolarisation fronts led to precipitating proton enhancements into the ionosphere with integral energy fluxes of the order of 0.1 µW m$^{-2}$. This corresponds to $6\times10^4$ keV cm$^{-2}$ s$^{-1}$. Assuming a uniform flux along all downwards directions as a rough approximation ($2\pi$ solid angle), this gives a directional integral energy flux of the order of $10^4$ keV cm$^{-2}$ s$^{-1}$ sr$^{-1}$, which is roughly three orders of magnitude below values obtained with Vlasiator when the global magnetotail reconfiguration is taking place (cf. Fig. 7b). A likely explanation is that the MHD simulation has low ion temperature $T_i$ values compared to our kinetic approach, leading to lower precipitation energy fluxes as these are proportional to $T_i\sqrt{T_i}$ in their approach (cf. eq. (1) in Ge et al., 2012). We note that the Vlasiator integral flux values are on the other hand in agreement with **the test-particle simulation results presented in their companion paper (Zhou et al., 2012b) with integral energy fluxes of 0.2–1 mW m$^{-2}$ (equivalent to $2\times10^7$–$10^8$ keV cm$^{-2}$ s$^{-1}$ sr$^{-1}$ with the same reasoning as above), and with** statistical patterns shown in Galand et al. (2001) (~0.1 mW m$^{-2}$). One strength of evaluating the proton precipitation parameters using a kinetic model is that not only integral energy fluxes can be calculated, but also differential fluxes, which may enable more detailed future studies of proton precipitation and its link to global magnetospheric dynamics. **Further, a test-particle approach is not fully self-consistent, in the sense that the electromagnetic fields affect the particle distributions but not vice versa. Therefore the test-particle approach does not fully describe, e.g., dynamics of reconnection-related precipitation.**"

– p. 20, l. 33: "The examination of precipitation bursts passing by virtual spacecraft S$_1$ between $t = 1920$ s and $t = 2080$ s suggests that these bursts are associated with dipolarising flux bundles originating from the vicinity of the stable X-line in the current sheet. **Field-aligned beams of plasma propagating earthwards associated to dipolarisation fronts have been observed and simulated not only in the central plasma sheet, but also in the plasma sheet boundary layer (PSBL) (Zhou et al., 2012a). In this Vlasiator run, there is unfortunately no cell located in the PSBL where the full VDF is saved at each time step to make a comparison with results from Zhou et al. (2012a), hence we focus on proton precipitation originating from the central plasma sheet.**"

There are two model features that require some explanations/discussions: Figure 3: what is a local temperature minimum around S1 position? Temperatures earthward and tailward from S1 are higher than in the S1 location. Is there any analogy of such temperature minimum in the statistical spacecraft observations? As I know, the temperature profile along the magnetotail is generally monotonous (see, e.g., 10.1029/2008JA013849, 10.1002/2016JA023710.)

First of all, something to note is that the shown temperature is the temperature of an isotropic plasma which would have the same total pressure as the simulated distribution, and thus does not represent the temperature of the bulk plasma but rather the measured effect of a combined bulk plasma and fast additional flow.

The local temperature minimum seen in Fig. 3 comes from the fact that hot plasma associated with the precipitating protons originates from the current sheet region, which leads to temperature enhancements propagating earthwards in terms of L shells. At the time step chosen for Fig. 3 (1800 s), a stream of hot ($T = 2\times10^7$ K) plasma coming from the transition region is reaching S$_1$, leading to a local enhancement of the proton temperature compared to its background value at S$_1$.

It is difficult to compare our results to those shown in the two suggested references, as in 10.1029/2008JA013849 the statistical observations are obtained with slow flows ($V_\perp < 150$ km/s) whereas in the situation shown in our Fig. 3 $V_\perp$ is essentially greater than 200 km/s for X $< -7$ $R_E$. As for the $T_i$ profile along the magnetotail shown in 10.1002/2016JA023710, it consists of only three points with X between $-10$ $R_E$ and $-30$ $R_E$, meaning that (i) the spatial resolution is too coarse

to capture such small-scale variations as that pointed in our Figure 3, and (ii) measurements at $X > -10\ R_E$ are not available.

We therefore did not refer to the above two papers in the revised manuscript; however, we have added the following discussion on the plasma temperature shown in Fig. 3:

– p. 9, l. 20: "The main panel (a) shows the proton temperature in the nightside part of the noon-midnight plane at time step t = 1800 s, with magnetic field lines drawn in black. **It should be stressed that this temperature is that of an isotropic plasma which would have the same total pressure as the simulated distribution, thus it does not represent the temperature of the bulk plasma but rather the measured effect of a combined bulk plasma and potential fast additional flows. The effect of such a combination can be seen near $S_1$, as a stream of hot $(T \approx 20\ MK)$ plasma coming from the transition region is reaching the virtual spacecraft, leading to a local enhancement of the proton temperature compared to its background value at $S_1$.**"

Figure 4: some of shown distributions are definitely unstable (they contain ion beams with positive slobs along the parallel velocity direction). Thus, some discussion is needed to explain if these instabilities are too slow to influence ion distributions or they are simply suppressed in the numerical calculations.

Thank you for raising this issue. In Vlasiator, instabilities are resolved in the numerical calculations provided their wavelengths are larger than the grid resolution in ordinary space. According to Gary (1989, 10.1007/BF00196632; see especially Table II p. 385), there are several instabilities which can arise from interactions between a proton beam and a proton core population in a plasma.

The ion/ion right-hand resonant instability ("magnetosonic" or "fast MHD"), of much lower frequency than the ion gyrofrequency, can develop when the field-aligned drift velocity $V_0$ of the protons is greater than the Alfvén speed $V_A$, and when the ratio between beam ($n_b$) and core ($n_c$) densities is "very small". If we consider the distribution shown in Fig. 4a of our manuscript, we have $V_0 \sim 1000$ km/s, $V_A = 4379$ km/s, $n_b = 1.1e4$ m$^{-3}$ and $n_c = 1.2e5$ m$^{-3}$, i.e., $V_0/V_A \sim 0.2$ and $n_b/n_c = 8.6e-2$. Since the condition on the velocities is not fulfilled, it is unlikely that this instability grows at the location where the VDF is observed.

The ion/ion left-hand instability also requires $V_A < V_0$, and in addition it requires that the ion beam to be hot, i.e., with a thermal velocity greater than the beam drift velocity. This instability is therefore not expected to grow in the considered situation of Fig. 4a.

The ion/ion nonresonant instability ("firehose") can develop if $V_A << V_0$. Hence, it cannot grow in this situation either.

In conclusion, in the $V_A > V_0$ regime which characterises the plasma in the vicinity of virtual spacecraft $S_1$ and $S_2$, we do not expect that the instabilities listed above can grow fast enough to significantly affect the ion distributions on the time scale needed for precipitating protons (i.e., essentially the field-aligned beam) to reach the inner boundary or even the ionosphere.

We have included a shorter version of this discussion in the revised manuscript as follows:

– p. 12, l. 1: "The crescent-shaped beam partly in the bounce loss cone shown in panel (a) is associated with a precipitating differential flux narrow in energies (panel e), peaking around 4–5 keV with values close to $10^4$ proton cm$^{-2}$ s$^{-1}$ sr$^{-1}$ eV$^{-1}$. **Although this distribution is intrinsically unstable, the development of instabilities (magnetosonic, ion-ion left-hand, firehose) requires the conditions that the field-aligned drift velocity $V_0$ of the protons be greater than the Alfvén speed $V_A$, and that the ratio between beam ($n_b$) and core ($n_c$) densities be "very small" (Gary, 1991). While the distribution shown in panel (a) has $n_b/n_c = 0.086$, it exhibits $V_0 \sim 1000$ km s$^{-1}$ and $V_A = 4379$ km s$^{-1}$. Since the criterion on velocities is not verified, we do not expect that the instabilities listed above can grow fast enough to significantly affect the ion distributions on the time scale needed for precipitating protons (i.e., essentially the field-aligned beam) to reach the inner boundary or even the ionosphere.**"

**Manuscript angeo-2019-59: "Hybrid-Vlasov modelling of nightside auroral proton precipitation during southward interplanetary magnetic field conditions" by M. Grandin et al.**

**Response to Reviewer #2**

We thank the Reviewer for their careful analysis of our manuscript as well as their constructive comments and suggestions. We provide our responses to the comments in blue, while the original text written by the Reviewer is shown in black.

The manuscript deals with the nightside proton precipitation, focusing on the perspective of the magnetosphere, by using Vlasiator, a global kinetic hybrid simulation of the near-Earth environment. The authors found a good agreement in terms of differential number fluxes and energies, compared to the empirical Hardy model. The proton precipitations are observed as burst events and, in some cases, they can be traced back to the current sheet in the magnetotail and associated with depolarising flux bundles.

The paper brings interesting results and can be acceptable for publication after some clarifications.

1. In my opinion, the authors should justify the resolutions chosen for the simulation in terms of characteristic plasma quantities. In particular, in order to comment on the physical space resolution, what is the value for the ion inertial length? or Larmor radius? Since the proton precipitation is produced by the magnetic reconnection in the magnetotail, how thin is the current sheet? And, in a similar way, the authors should comment on the velocity space resolution. In this respect, what is the value for the Alfvén speed?

Thank you for this nice suggestion. We are indeed happy to provide these values and put them in perspective with the ordinary and velocity space resolutions. We can for instance provide such values in the solar wind (input) and in the transition region (at $X = -10\ R_E$, $Y = Z = 0$) at $t = 1800$ s (Fig. 2).

| Location | Solar wind | Transition region |
|---|---|---|
| – ion inertial length: | $\lambda_p = 228$ km | $\lambda_p = 906$ km |
| – ion Larmor radius: | $r_L = 214$ km | $r_L = 681$ km |
| – Alfvén speed: | $V_A = 109$ km/s | $V_A = 1596$ km/s |

The ordinary space resolution of 300 km in this run is therefore sufficient to resolve most of the proton kinetics, as was confirmed in a dedicated study by Pfau-Kempf et al. (2018, 10.3389/fphy.2018.00044).

This discussion has been added to the revised manuscript as follows:

– p. 5, l. 6: "**In the solar wind, the ion inertial length is $\lambda_p = 228$ km, and the proton Larmor radius is $r_L = 214$ km. A dedicated study by Pfau-Kempf et al. (2018) showed that the ordinary space resolution of 300 km is sufficient to resolve most of the proton kinetics in such conditions. Besides, the Alfvén speed is $V_A = 109$ km s$^{-1}$, which is significantly greater than the velocity space resolution of 30 km s$^{-1}$. Similarly, the ordinary space and velocity space resolutions are sufficient to resolve the ion kinetics in the transition region between stretched and dipole-like magnetic field lines in the magnetotail, where $\lambda_p = 906$ km, $r_L = 681$ km and $V_A = 1596$ km s$^{-1}$.**"

The plasma sheet thickness was evaluated during this same run by Juusola et al. (2018b, 10.5194/angeo-36-1027-2018) and is shown in Fig. 6 of their paper. In the region of interest for our study ($X > -20\ R_E$), Juusola et al. find that the plasma sheet thickness essentially lies within 0.1–0.2 $R_E$ (i.e., just a few simulation cells). This discussion has been added as follows:

– p. 6, l. 6: "**Juusola et al. (2018b) studied the plasma sheet during this same run and evaluated its thickness to lie essentially within 0.1–0.2 $R_E$ (i.e., just a few simulation cells) in the region of interest for our study ($X > -20\ R_E$).**"

2. In the inner boundary, protons are described with a static Maxwellian VDF. This means that here kinetic effects are neglected. However, the observed VDFs close to the inner boundary, for example in S2 (Fig.3), are strongly non Maxwellian during the precipitation. I am wondering if the imposed sharp change in the VDF could influence the results. Could the authors comment on this point?

The imposed static Maxwellian VDF at the inner boundary mostly affects the neighbouring cells through the calculation of translation and acceleration terms in the Vlasov equation. Since $S_2$ is located about 15 cells away from the inner boundary, its stencil used for acceleration and translation do not reach the inner boundary. Given that, in addition, the plasma flows from $S_2$ towards the inner boundary, potential diffusion effects are likely negligible in comparison with the bulk flow.

We have added a short discussion on this aspect as follows:

– p. 11, l. 3: "**The imposed static Maxwellian VDF at the inner boundary can affect the neighbouring cells through the calculation of translation and acceleration terms in the Vlasov equation, but such effects vanish rapidly with distance to the boundary, and the plasma at $S_2$ is unlikely to be affected by the boundary condition.**"

3. One of the hypotheses used to evaluate the directional differential particle flux is that 'protons remain attached to a given magnetic flux tube'. However, during a magnetic reconnection event this is not exactly true. Although the locations in the magnetosphere are chosen far from the X-point, I am wondering if the change in the magnetic topology can have effects also at these points for the analysis.

Since the locations of virtual spacecraft $S_1$ and $S_2$ are on closed field lines which do not further reconnect and are convected earthwards, there is no risk that the precipitating protons observed at $S_1$ (or $S_2$) are affected by magnetic reconnection before they reach the ionosphere. Hence, there should be no concerns regarding the validity of the hypothesis that the plasma observed at the virtual spacecraft remains attached to its magnetic flux tube until precipitating protons reach the ionosphere.

We have mentioned this aspect in the following addition to the revised manuscript:

– p. 13, l. 4: "**Furthermore, given the location of $S_1$ and $S_2$ on closed field lines which are being convected earthwards, there is no risk that the precipitating protons observed at $S_1$ (or $S_2$) are affected by processes such as magnetic reconnection before they reach the ionosphere.**"

4. During the phase 2, the orbit of the NASA-MMS mission was chosen to spend time on the night side of the Earth's magnetosphere. Did the authors check if there are any datasets able to support their results?

Comparing our results with MMS data would indeed be interesting; however, we feel that this task would be beyond the scope of the present study, whose main aim is twofold: (i) present the methodology to evaluate proton precipitation in Vlasiator simulations, and (ii) discuss the nightside proton precipitation from a global perspective during a simulated event with southward IMF. A comparison of those results with observations could therefore be carried out in a follow-up study.

We hence mention this idea for future work as follows:

[revised manuscript text omitted]